# Photooxidation driven formation of Fe-Au linked ferrocene-based single-molecule junctions

Woojung Lee [1], Liang Li [1], María Camarasa-Gómez [2], Daniel Hernangómez-Pérez [2], Xavier Roy [1], Ferdinand Evers [2] ✉, Michael S. Inkpen[3] ✉ & Latha Venkataraman [1,4] ✉

Metal-metal contacts, though not yet widely realized, may provide exciting opportunities to serve as tunable and functional interfaces in single-molecule devices. One of the simplest components which might facilitate such binding interactions is the ferrocene group. Notably, direct bonds between the ferrocene iron center and metals such as Pd or Co have been demonstrated in molecular complexes comprising coordinating ligands attached to the cyclopentadienyl rings. Here, we demonstrate that ferrocene-based single-molecule devices with Fe-Au interfacial contact geometries form at room temperature in the absence of supporting coordinating ligands. Applying a photoredox reaction, we propose that ferrocene only functions effectively as a contact group when oxidized, binding to gold through a formal $Fe^{3+}$ center. This observation is further supported by a series of control measurements and density functional theory calculations. Our findings extend the scope of junction contact chemistries beyond those involving main group elements, lay the foundation for light switchable ferrocene-based single-molecule devices, and highlight new potential mechanistic function(s) of unsubstituted ferrocenium groups in synthetic processes.

Ferrocene is a prototypical organometallic compound that comprises a single iron atom sandwiched between two cyclopentadienyl rings ($FeCp_2$). Since its discovery in the mid-20th century[1,2], ferrocene and its derivatives have enjoyed extensive utilization due to their stability under ambient conditions, facile synthetic modification, and well-defined reversible electrochemistry[3–7]. These properties have been exploited in seminal works related to electron transfer and transport, for example in mixed-valence complexes[8], thin organic films[9], or multi-molecular devices with rectification ratios on the order of $10^5$ [10]. Ferrocene derivatives have also been used as molecular wires where the ferrocene is implicated as a contact for nanoscale Au electrodes[11–15]. These reports are primarily at low temperatures where ferrocene can

act as a linker through the Cp ring due to van der Waals interactions with the Au electrode[12,15]. Direct bond formation between a ferrocene iron center and other metals has been achieved in complexes with ancillary metal-binding ligands attached to the Cp ring, where a dative bond, denoted as Fe→M, is formed[16–19]. Due to the filled frontier orbital of ferrocene[20,21], the Fe→M bond has been observed primarily with closed-shell metals having no unpaired electrons such as Ru(II), Pd(II), and Pt(II)[22,23] although there are a few exceptions[24].

Here, we leverage the electrochemical advantages of ferrocene derivatives as demonstrated in recent research exploring their application as photocatalysts[25–29] to create a light-controlled ferrocene-based single-molecule device using the scanning tunneling microscope-based

[1]Department of Chemistry, Columbia University, New York, NY 10027, USA. [2]Institute of Theoretical Physics, University of Regensburg, 93040 Regensburg, Germany. [3]Department of Chemistry, University of Southern California, Los Angeles, CA 90089, USA. [4]Department of Applied Physics and Applied Mathematics, Columbia University, New York, NY 10027, USA. ✉e-mail: ferdinand.evers@physik.uni-regensburg.de; inkpen@usc.edu; lv2117@columbia.edu

break junction (STM-BJ) technique, as depicted in Fig. 1a. We apply photo-induced ferrocene oxidation[25] to form Fe-M bonded junctions between a ferrocene iron center and an open-shell metal, $Au^0$ (an undercoordinated Au atom on the electrode) without ligand-support. A series of control measurements and ab initio-based quantum transport calculations corroborate the observed results and confirm the feasibility of our proposed ferrocene-coordinated structural motif. Our work thus supports an Fe-Au bond can be formed by manipulating the oxidation state of ferrocene using light, creating single-molecule devices linked through a metal-metal interface at room temperature.

## Results and discussion
### Photoredox measurements
We perform single-molecule conductance measurements using a STM-BJ technique (see methods for details) in the presence of a 405 nm laser. We form a single-molecule junction of **1**, a ferrocene derivative with two thioanisole groups in the presence of a photoredox agent as illustrated in Fig. 1. Synthetic details are provided in Supplementary Method. A solution containing **1** and bis(4-tert-butylphenyl)iodonium

hexafluorophosphate ($[R_2I]^+[PF_6]^-$) in a 9:1 mole ratio was prepared in propylene carbonate (PC) at a concentration of 1 mM. Under irradiation by a 405 nm-laser at an intensity of around 100 mW cm$^{-2}$, **1** undergoes oxidation in solution. As illustrated in Fig. 1c, STM-BJ measurements at a tip bias of 100 mV with the laser irradiation results in the formation of a molecular junctions with a conductance of $-1 \times 10^{-3} G_0$. Since the applied tip bias is lower than the redox potential of **1** (280 mV, see Supplementary Note 1 and Supplementary Fig. 1), the oxidized ferrocene complex in solution is reduced back to its neutral form when the laser is turned off, leading to the formation of molecular junctions exhibiting a distinct conductance of $-6 \times 10^{-5} G_0$. The formation of molecular junctions is not inhibited by any interaction between $[1]^+$ and $[PF_6]^-$ under the photoredox conditions. (Supplementary Fig. 2) We propose that the molecular junctions of **1** have distinct geometries **1H** and **1L** depending on whether the laser is on or off as illustrated in Fig. 1b, with the **1L** junction linked solely through the terminal thioanisole groups and the **1H** junction linked through an Fe-Au bond. Through the photoredox reactions, we can manipulate the charge states of ferrocene-based single-molecule devices, enabling

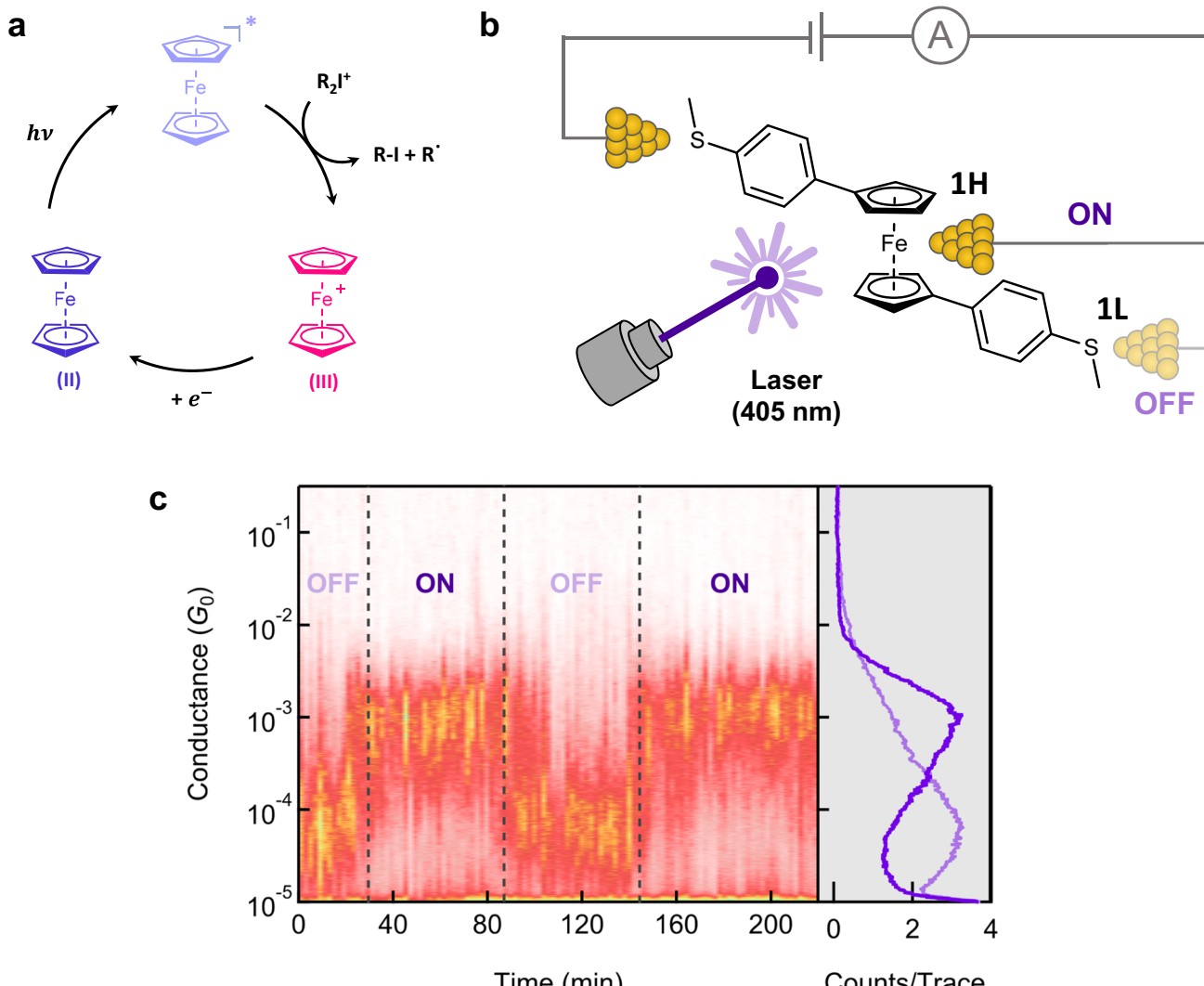

**Fig. 1 | Schematic of photoredox reaction studied, the scanning tunneling microscope-based break junction (STM-BJ) measurement, and conductance results. a** Mechanism of photoredox reaction for ferrocene derivatives. $R_2I^+$ is an iodonium salt (R: 4-tert-butylphenyl; counterion: $[PF_6]^-$). **b** Schematic representation of two distinct single-molecule junction geometries formed with **1** between two Au electrodes during scanning tunneling microscope-based break junction (STM-BJ) measurements. **1H** and **1L** denote two distinct junction geometries. **c** Left: Time-resolved conductance histograms of **1** measured at a 100 mV bias with the laser turned on or off (as indicated in the figure). Histograms are created by compiling consecutive sets of 100 conductance-distance traces. Right: Total one-dimensional (1D) conductance histogram of **1** showing conductance changes when with the 405 nm-laser on (dark purple) or off (light purple).

control over the geometries of interfacial contact and, consequently, the resulting junction conductance.

## Electrochemical redox measurements

We next performed single-molecule conductance measurements of **1** using a 50 μM solution in PC without the photochemical oxidants. In Fig. 2b, we plot one-dimensional (1D) conductance histograms for **1** obtained at three tip biases: −450 mV, 100 mV, and 450 mV, two below the oxidation potential (280 mV) and one above (see Supplementary Note 1 and Supplementary Fig. 1). At these biases, we observe a clear conductance peak at ~ $6 \times 10^{-5} G_0$. However, at a bias of 450 mV, when **1** is in an oxidized state, junctions exhibit a much higher conductance (~$1 \times 10^{-3} G_0$), which aligns with the findings from our photoredox experiments. To confirm that these peaks arise from conductance plateaus formed when a single-molecule is held between the tip and the substrate, we create two-dimensional (2D) conductance displacement histograms and show these in Supplementary Fig. 3. We obtain a plateau length for **1L** of ~ 5 Å in its neutral state (100 mV and −450 mV), while the oxidized state of **1H** (450 mV) exhibits a much shorter plateau length of ~2 Å. We note that Au electrodes undergo relaxation and reorganization upon rupture of the Au-Au contact, resulting in a difference between the plateau length and actual molecular junction length, known as the snapback distance[30–32]. The reported snapback distances for various molecular structures and solvents are around 5–8 Å[33–36]. After taking into account this snapback distance, the shorter plateau length for the oxidized molecule is consistent with the distance between sulfur and iron in **1**, ~7.4 Å[33–35].

In polar solvents, we can alter the charge state of the molecule by changing the junction bias; however, in non-polar solvents without electrolytes, it is not possible to tune the charge state of the molecule by altering the tip bias. To test how the solvent impacts conductance data, we measure **1** in tetradecane (TD), a non-polar solvent, and find that we do not observe the **1H** junction geometry even at a bias as high as 700 mV. We do however observe the **1L** junctions as shown in Supplementary Fig. 4. As a third control experiment, we add a chemical oxidant to a measurement of **1** at 100 mV in PC and observe high-conducting junctions analogous to those observed in experiments using light and the photoredox agent, and those obtained at a positive bias higher than the redox potential threshold of 280 mV (Supplementary Fig. 5). Moreover, employing an extra gate electrode to control the oxidation state of **1**, while keeping the tip-substrate bias below the redox potential, leads to the formation of two distinct junction geometries. (Supplementary Fig. 6) These observations confirm that **1** forms distinct junction geometries **1H** and **1L** in its oxidized and neutral states, respectively.

Next, we perform STM-BJ measurements of **2** from PC solutions. **2** is analogous to **1** but has only one thioanisole group and thus can form junctions only if the Fc group binds to the electrode. We work at an applied tip biases below (50 mV) and above (450 mV, 550 mV) the redox threshold determined from in situ CV measurements (Supplementary Fig. 1). Figure 2c shows 1D conductance histograms of all the measured traces where we see a clear conductance peak (at ~5 × $10^{-4} G_0$) only for the measurements at 450 mV and 550 mV, i.e., when the molecule is oxidized. The molecular junction length of **2** is similar to that of **1H**, around 2 Å. At 50 mV bias, no peak is obtained and the 1D

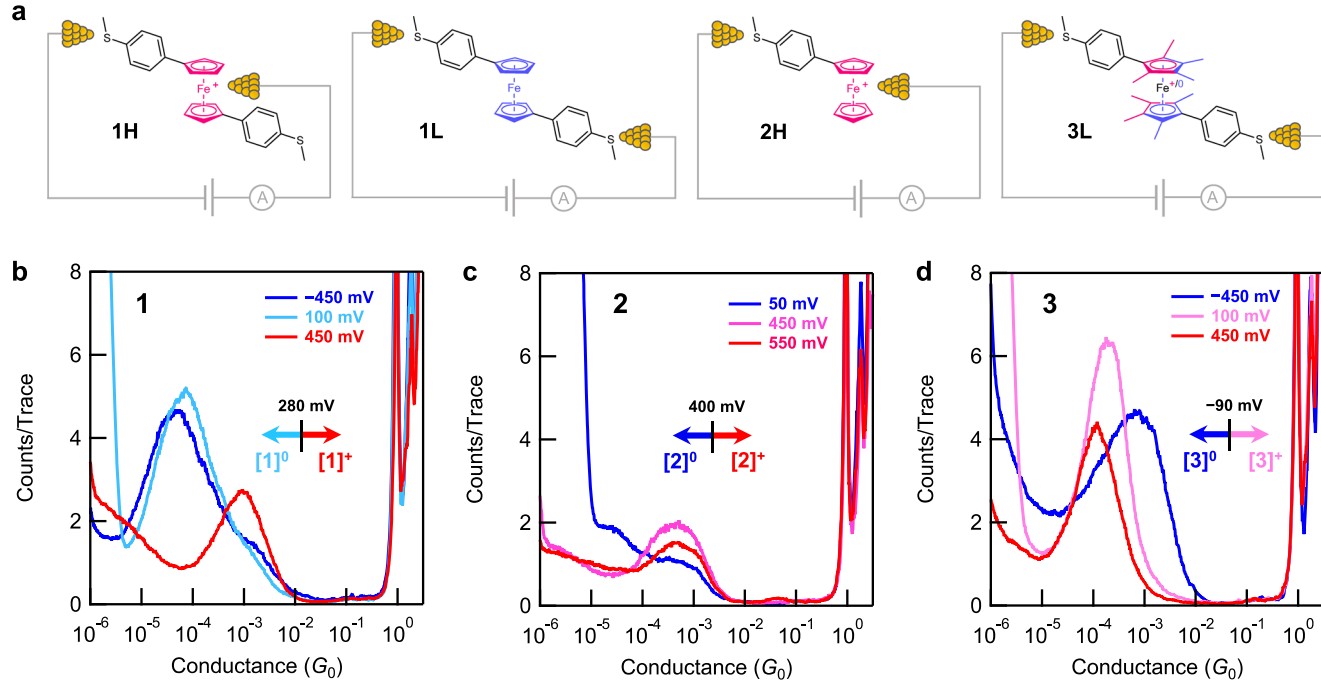

**Fig. 2 | Geometries of ferrocene derivative-based molecular junctions and conductance histograms at different bias voltages. a** Chemical structures of **1**, **2**, and **3**, and their experimentally accessible junction geometries; **H** and **L** denote 'high-conducting' and 'low-conducting' junction geometries of each derivative, respectively. The red color and blue color represent the oxidized (1+) and reduced (neutral) states of the ferrocene complexes, comprising formal $Fe^{3+}$ and $Fe^{2+}$ centers, respectively. **2** forms a junction (**2H**) only in its oxidized state. **3** forms only the low-conducting junction geometries in either oxidized or neutral states. **b** Overlaid one-dimensional (1D) conductance histograms of **1** measured at −450 mV (dark blue), 100 mV (light blue), and 450 mV (red) in propylene carbonate (PC). The redox potential of **1** is determined as 280 mV (Supplementary Fig. 1). This indicates that the analyte population close to the junction is predominantly composed of the oxidized state ($[1]^+$) above 280 mV and the neutral form ($[1]^0$) below 280 mV. **c** Overlaid 1D conductance histograms of **2** measured using a tip bias of 50 mV (dark blue), 450 mV (pink) and 550 mV (red) in PC. The redox potential of **2** determined from in situ CV is 400 mV. **d** Overlaid 1D conductance histograms of **3** measured at −450 mV (dark blue), 100 mV (pink), and 450 mV (red) in PC. The redox potential of **3** determined from in situ CV is −90 mV. The red and blue color schemes denote oxidized and neutral states, respectively. Each histogram is compiled of over 3,000 traces measured consecutively. Two-dimensional (2D) conductance-displacement histograms for these data are shown in Supplementary Fig. 3.

and 2D conductance histograms and the data match those of measurements in solvent alone (Supplementary Fig. 3b). As a control, we note that measurements of ferrocene without any thioanisole do not show any conductance feature at applied tip biases below or above its redox potential (Supplementary Fig. 7). These observations indicate that **2** forms a molecular junction in only its oxidized state, forming the **2H** junction geometry, and that the oxidized ferrocene unit can only serve as a contact group for one gold electrode. We therefore conclude that the oxidized states of **1** and **2** form junctions with high conductance (**1H** and **2H**) via an Fe-Au bond, while the neutral state of **1** forms the low conductance geometry (**1L**) bound by the two SMe linkers.

To verify our hypothesis, we control the formation of the Fe-Au bond through chemical design with derivative **3**, which has four methyl groups in addition to the thioanisole linker on each Cp ring. As shown in Fig. 2d and Supplementary Fig. 3, **3** does not form high conducting junctions, but we clearly see longer and lower conductance plateaus indicating that **3L** junctions are the only ones formed. We conclude that the Fe atom is not accessible to the Au electrode due to the steric bulk of methyl groups on the Cp ring (Supplementary Fig. 10). Note that the oxidation potential of **3** (−90 mV) is much lower than that of **1** (280 mV) thus we need to work at a large negative bias to measure the neutral form, while we observe the oxidized molecule at both 100 mV and 450 mV biases. These experimental results lead us to infer that the interfacial contact in **1H** differs from that in **1L**, which solely involves Au-SMe donor-acceptor bonds. Furthermore, the introduction of steric hindrance, primarily restricting access to the Fe atom, suggests that the formation of **1H** and **2H** junctions is attributed to the Fe-Au bond, rather than other bonds like Au-Cp. We note that the molecular junction conductance of **3** is higher in its neutral state (−450 mV) than in the oxidized state, as confirmed by transmission calculations detailed below.

We next validate the formation of the Fe-Au bond again using a modified measurement method where we first pull the Au-Au contact apart at a bias of 450 mV or 100 mV at a rate of 20 nm·s⁻¹ for 150 ms, then hold the junction for 150 ms, and then pull the junction apart for an additional 200 ms to fully break the contact before restarting the measurement. When holding the junction, we drop the bias to 100 mV in the central 50 ms portion. As discussed above, we can trap either **1H** or **1L** by choosing the initial bias to be either 450 mV or 100 mV, respectively. When using an initial bias of 450 mV, we hold the **1H** junction and can determine its conductance at 100 mV. We compile all traces that start and end with a molecular junction during the hold into 2D conductance-time histograms and show these in Fig. 3. The most probable conductance of **1H** and **1L** junctions at 100 mV differ by a factor of ~3. The conductance of each junction geometry at 450 mV is also shown in Supplementary Fig. 8. By contrast, an analogous measurement with **3** shows no difference based on the initial bias during the hold segment (Supplementary Fig. 9). This confirms that the different conductance and plateau lengths we observe for junctions formed from **1** at oxidizing or reducing tip biases correspond to distinct electrode-molecule contacts. We are not simply forming and measuring junctions of the same geometry in different oxidation states.

## Flicker noise measurements

To corroborate the formation of the Fe-Au bond, we conducted STM-BJ based flicker noise measurements[37]. During these measurement, we hold the molecular junctions of **1** for 150 ms at biases of 150 mV and 450 mV to form **1L** and **1H** junctions, respectively. We obtain a discrete Fourier transform of the measured conductance during the hold period, and square it to determine the conductance noise power spectral density (PSD) for each junction as shown in Fig. 4a (see "Methods" section for details). Since flicker noise in single-molecule junctions depends on the molecule-electrode coupling, the relation between flicker noise power and molecular conductance (G) can indicate the type of coupling[37]. Specifically, flicker noise shows a power-law dependence on conductance: noise power is proportional to $G^n$.

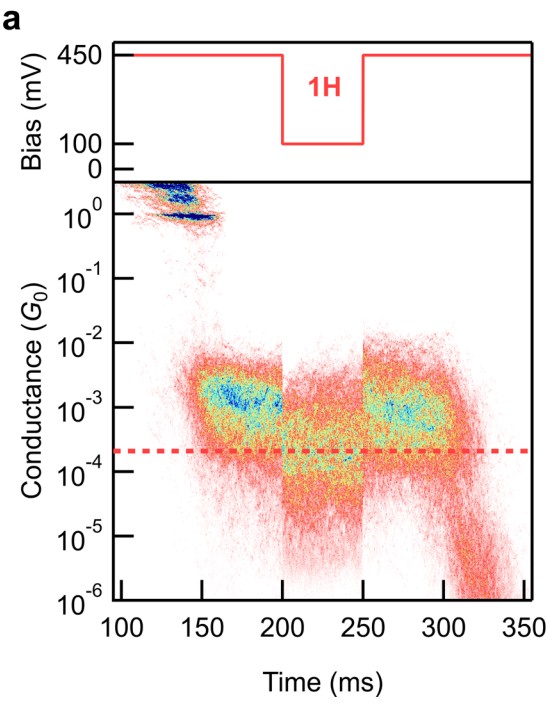
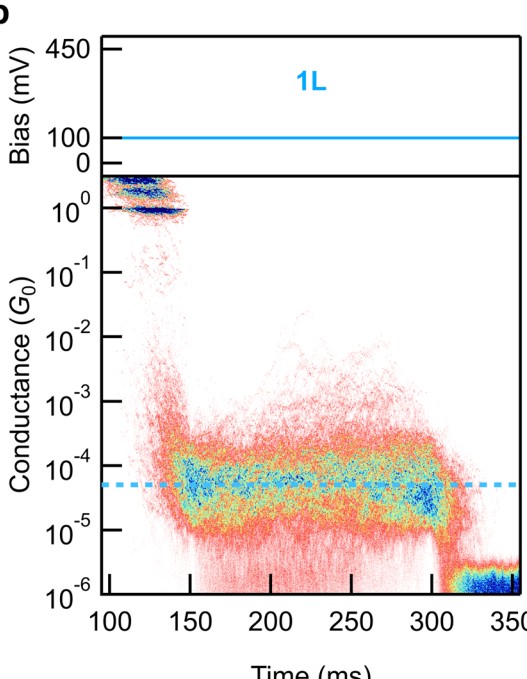
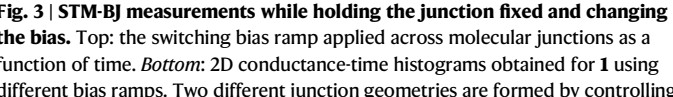

**Fig. 3 | STM-BJ measurements while holding the junction fixed and changing the bias.** Top: the switching bias ramp applied across molecular junctions as a function of time. *Bottom*: 2D conductance-time histograms obtained for **1** using different bias ramps. Two different junction geometries are formed by controlling the oxidation state of **1** with the applied voltage: (**a**) **1H** at 450 mV and (**b**) **1L** at 100 mV. The most probable measured conductance of both geometries at 100 mV is indicated with the dashed line.

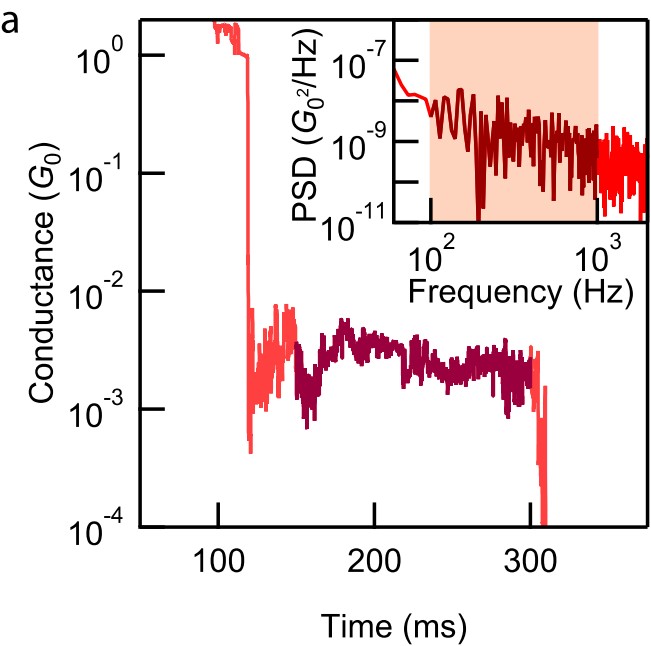

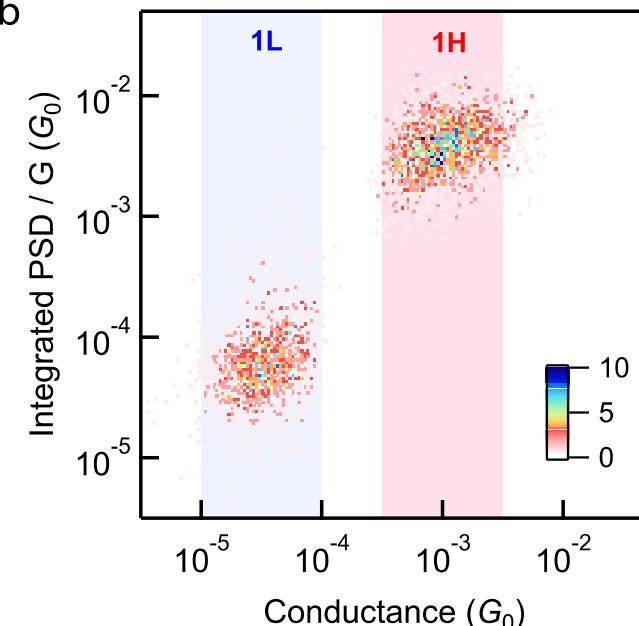

**Fig. 4 | Flicker noise measurements. a** Single conductance versus time trace for a flicker noise measurement of **1** at 450 mV. During the measurement, the molecular junction is held for 150 ms (dark red region). Inset: the noise power spectral density (PSD) obtained by taking the modulus square of the discrete Fourier transform of the hold segment. **b** 2D histogram of integrated normalized flicker noise power *versus* average junction conductance for **1L** (at 150 mV) and **1H** (at 450 mV). Conductance regions corresponding to **1L** and **1H** are indicated with blue and red sections respectively. The exponents describing the relationship between integrated flicker noise and conductance are as follows; 1.28 for **1L** and 1.30 for **1H**.

Through-bond coupled junctions have a characteristic $n$ around 1 while through-space coupled junctions have an $n$ close to 2[37]. The 2D histograms of normalized flicker noise power integrated over a frequency ranging from 100–1000 Hz *versus* average junction conductance for **1L** and **1H** are shown in Fig. 4b. The exponent $n$ determined from the noise spectra for the **1L** and **1H** geometries were determined to be the same, i.e., 1.28 and 1.30. We ascribe the deviation from $n = 1$ to quantum interference arising from the rotation of Cp rings which can lead to changes in junction conductance[14]. Importantly, the exponent close to $n = 1$ indicates strongly that the coupling in **1H** is a through-bond coupling and not a through-space coupling that involves van der Waals interactions between the Cp ring and the Au electrode. Therefore, we conclude that the coupling to the Au electrodes is through an Fe-Au bond when the junction conductance is high. We will discuss this further in the following calculation section.

## First-principles calculations

We computationally rationalize that oxidized ferrocene derivatives form a ferrocene-gold (Fc-Au) contacts through the Fe-Au bond based on density functional theory (DFT) calculations using the FHI-aims software (see Methods for details)[38–40]. First, we study the electronic interactions of the Fc-Au contact. To simulate the STM-BJ measurement, we relax the geometry of a ferrocene molecule near a fixed Au electrode (Au$_{22}$ cluster) and determine the binding energy including van der Waals (vdW) interactions (see Supplementary Note 3 and Supplementary Fig. 11)[41]. For the molecule bound through an Fe-Au bond, we obtain a binding energy of around 0.80 eV (vdW contribution is 0.38 eV) while for Fc that has an π orbital-Au interaction between the Cp ring and the Au cluster, the binding energy is 0.29 eV (vdW contribution is 0.22 eV). This indicates that the Fc is unlikely to form a junction at room temperature unless an Fe-Au bond is formed. Note that experiments that find that Fc adsorbed on Au surfaces through vdW interactions desorbs at temperatures above 250 K[15,42].

In order to relate the measured conductance to the **H** and **L** molecular junction geometries, we carried out electron transmission calculations of the junctions with the neutral and oxidized ferrocene derivatives using the non-equilibrium Green's function formalism (NEGF). We employ DFT as implemented in FHI-aims and the AITRANSS transport package for the NEGF calculations (see Methods for details)[38–40]. Simplifying our analysis, we focus on comparing **1L** and **2H**, assuming the electron tunneling properties of the **2H** junction to be analogous to the **1H** junction, due to their same electron transmission pathway. To calculate the transmission across the **2H** junction, we first relax the isolated molecule with one Au atom attached to the SMe linker and one attached to the Fe atom. Next, we add two Au electrodes by attaching Au clusters to the Au atoms at both ends (Fig. 5b). Since each Au atom has an unpaired electron, we add an even-number Au$_{22}$ cluster at the Fe-Au side to provide net antiparallel spin configuration and an odd-number Au$_{21}$ cluster at the S-Au side. The resulting net charge on **2** within the junction is +0.773, indicating that the **2H** junction is in an oxidized state. Similarly, for the **1L** junction, we first relax the molecule with two Au atoms appended at each thioanisole group and then add an Au$_{21}$ cluster at each end to model the **1L** junction (Fig. 5a). After the junction geometries of **1L** and **2H** are relaxed, we calculate the transmission functions for both junctions. These are shown in Fig. 5c.

The transmission of **1L** reaches 1 at the resonances corresponding to the highest occupied molecular orbital (HOMO) and lowest unoccupied molecular orbital (LUMO) as seen in Fig. 5c. This is because the frontier orbitals of **1L** are symmetrically coupled through the S-Au bonds (Supplementary Fig. 12). However, for **2H**, these resonances do not reach unit transmission. The frontier orbitals of **2H** have more weight on the Fe-Au bond rather than S-Au (Supplementary Fig. 13) and thus these orbitals are not symmetrically coupled to both electrodes[43]. Additionally, we note that the resonances in the transmission of **1L** are narrower than those of **2H**, consistent with the fact that the HOMO of ferrocene is poorly electronically coupled to the Cp rings in the long geometries[14].

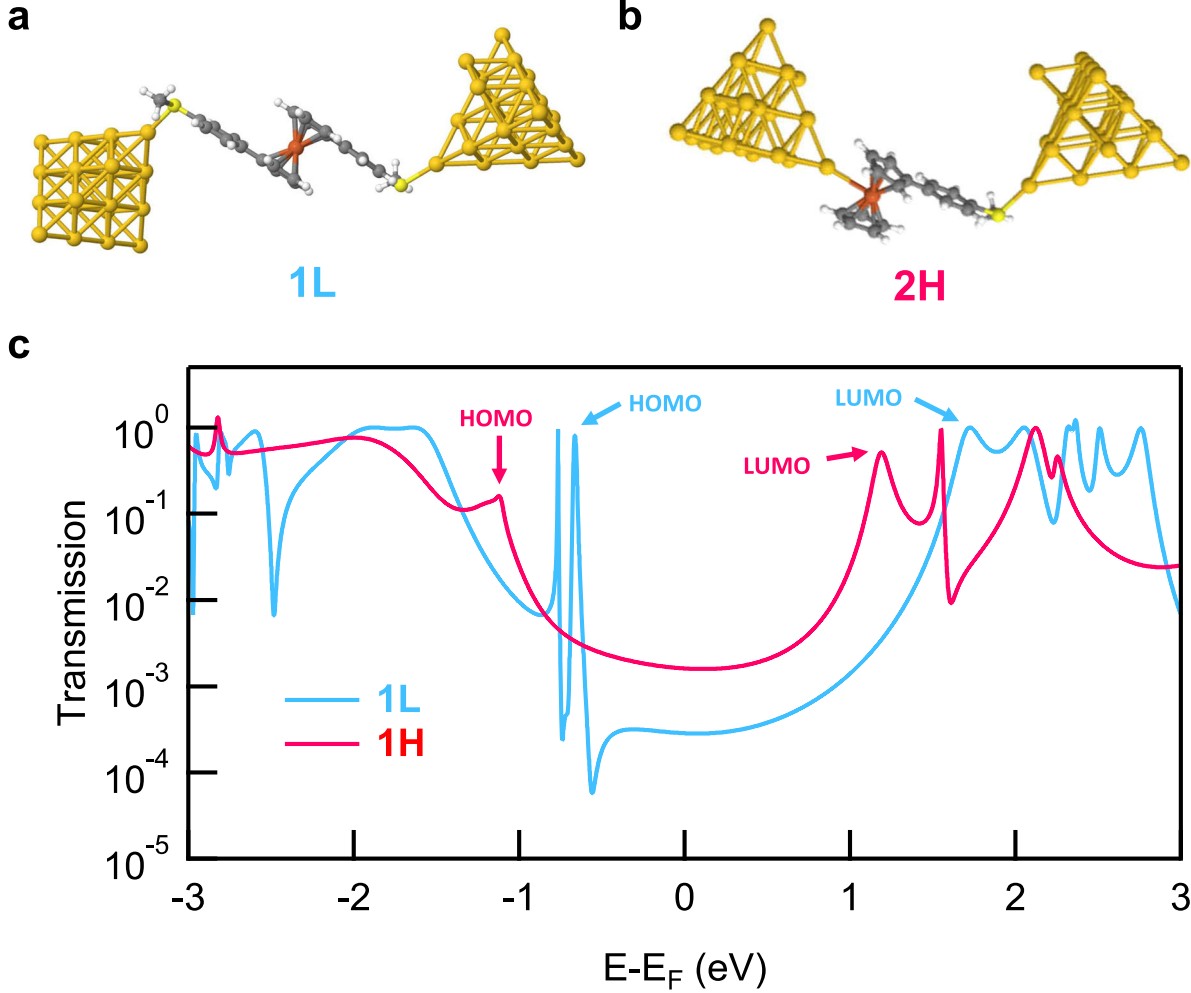

**Fig. 5 | Relaxed junction geometries and calculated transmission functions of 1L and 2H junctions.** The relaxed junction geometries for (**a**) **1L** and (**b**) **2H**. Dark gray, light gray, red, yellow, and gold spheres represent C, H, Fe, S, Au atoms, respectively. **c** Calculated transmission functions for **1L** and **2H**. Frontier orbitals resonance positions are indicated by the arrows.

Finally, we find the frontier orbitals of **2H** have opposite phase relations and thus interfere constructively leading to an increase in conductance around $E_F$ (Supplementary Fig. 13), whereas for **1L** the HOMO and HOMO-1 resonance interfere destructively as do the HOMO and LUMO resonances[44]. This decreases the conductance of **1L** at $E_F$ significantly when compared with **2H** (Fig. 5c). We note that the presence of an additional thioanisole group in **1** alters the direction of the electrode linked to Fe, resulting in a conductance difference between **1H** and **2H**, as shown in Fig. 2b. This observation is consistent with our previous findings[14] and further supported by the calculated transmission of **1H** in Supplementary Fig. 15. Taken together, the transmission calculations support the conductance trends of ferrocene derivatives determined experimentally in this work. We also show the results from transmission calculations of **3L** in Supplementary Fig. 16. These transmission calculations provide conductance trends that are consistent with our experimental data, supporting the hypothesis that the Fe-Au bond formation within ferrocene derivatives is available only in the oxidized state. Lastly, the spin density distribution of the Fe-Au contact indicates that the ferrocene iron center is more favorable for binding to the gold electrode than the Cp rings in its oxidized state (Supplementary Fig. 17).

In conclusion, we have introduced a photoredox reaction to create ferrocene-based single-molecule devices. We have demonstrated that ferrocene junctions are formed with a direct bond between a ferrocene Fe center and an Au electrode through a series of STM-BJ measurements and DFT-based calculations. The light-induced formation of such devices not only offers a systematic control method for manipulating single-molecule devices but also opens up avenues for the development of versatile and higher-conducting single-molecule junctions that were previously inaccessible with organic linkers. Although bond characteristics could not be studied using methods such as X-ray photoelectron spectroscopy, we hope that such studies will be carried out in subsequent studies.

## Methods

### Synthesis

**1** and **2** were prepared in a one-pot, multi-step process by extension of a previously reported approach[14]. First, a mixture of mono and 1,1'-dili-thioferrocene was prepared by reaction of ferrocene with $n$-butyl lithium in the presence of N,N,N',N'-tetramethylethylenediamine (a chelating diamine). These species were then subjected to transmetallation with zinc chloride to provide the corresponding organozinc compound. Subsequent Negishi cross-coupling with 4-bromothioanisole provided a mixture of **1** and **2** that could be separated using conventional chromatographic and crystallization techniques. **3** was prepared by a salt metathesis reaction between the thioanisole-appended lithium tetra-methylcyclopentadienide ligand and $FeCl_2$. Complete synthetic and characterization details are provided in the Supplementary Information (SI).

## STM break-junction measurements

Conductance measurements for ferrocene-based molecular junctions were done using a customized STM-BJ setup that is described in detail before[45]. A piezo actuator, used to drive a Au tip, is pushed to a Au substrate, forming a Au–Au contact with a conductance greater than $1G_0$ ($1G_0 = 2e^2/h$, the quantum of conductance). Subsequently, the Au tip is retracted rupturing the contact, allowing a molecule to bridge the gap between two Au electrodes, forming a single-molecule junction at a rate of 20 nm s$^{-1}$. A bias voltage is applied and the resulting current is measured to yield a conductance (= current/voltage) trace as a function of relative tip-substrate displacement at an acquisition rate of 40 kHz. This process is repeated thousands of times to obtain statistically reproducible data that is presented as conductance histograms. For the measurements reported here, we use solutions of the molecules in propylene carbonate (polar) and tetradecane (non-polar) solvents under ambient conditions at room temperatures. In polar solvents, the measurements generate capacitive and Faradaic background currents. The STM tip is therefore coated with wax to reduce the exposed surface area to under ~10 μm$^2$ [46]. Additionally, due to the large difference between the exposed surface areas of the coated tip and bare Au substrate, the voltage drop across the molecular junction is asymmetric, allowing in situ control of the redox state of the ferrocene derivatives[47]. The standard deviation calculated from the histogram peak positions generated from sets of 100 traces is 2–6%.

## Flicker noise measurement

Flicker noise measurements were conducted as described in detail before[37,48]. We first formed 1L and 1H junctions at 150 mV and 450 mV respectively, held the junction for 150 ms (as detailed above for the hold measurement), and measured the conductance with a 100 kHz sampling rate. At least 2000 traces that sustain a molecular junction were selected for the analysis. We obtained the average molecular conductance ($G$) and the normalized noise power (power spectral density (PSD)/$G$) from the hold period. The PSD was calculated from the square of the integral of a discrete Fourier transform of the measured conductance between 100 Hz and 1000 Hz. These frequency limits are constrained by the mechanical stability of STM-BJ setup (100 Hz) and the input noise of the current amplifier (1,000 Hz). Using the calculated parameters, we create 2D histograms of the normalized integrated noise power *versus* the average conductance. The relationship between noise power and molecular conductance is derived by determining the scaling exponent ($n$) for which PSD/$G^n$ and $G$ are not correlated.

## DFT calculations

The DFT calculations in this work were carried out using both closed-shell and open-shell Kohn-Sham formulation of DFT implemented in the FHI-aims software[39]. A non-empirical generalized gradient-corrected approximation (PBE) for the exchange-correlation functional was used[49]. Scalar relativistic corrections to the kinetic energy were considered at the atomic zeroth-order regular approximation level[50]. The Kohn-Sham states were represented using an all-electron basis set with tight computational settings (roughly equivalent to double zeta plus polarization quality for the molecular atoms and double zeta quality for the gold atom). For the open shell calculations, the orientations of spins in Fe and Au are set to be collinear. The calculation results were obtained using standard convergence criteria in the self-consistent field cycle for the difference in the spin density ($10^{-5}$ spins Å$^{-3}$) for both spin-up and spin-down, total energy ($10^{-6}$ eV), sum of Kohn-Sham eigenvalues ($10^{-4}$ eV) and forces ($10^{-4}$ eV Å$^{-1}$). The binding energies were obtained by the calculated energy difference between the bound geometry and the sum of the energies of the isolated ferrocene molecules and gold. The energy-dependent transmission functions were calculated using the non-equilibrium Green's function formalism with the transport package AITRANSS[51]. Additionally, the gold electrodes were modeled by tetrahedral clusters of Au atoms each with interatomic distance of 2.88 Å. The self-energy of the Au reservoirs is a local and energy-independent (Markovian) function modeled by the matrix $\Sigma(\mathbf{r}, \mathbf{r}') = i\eta(\mathbf{r})\delta(\mathbf{r}\text{-}\mathbf{r}')$, where $\delta(\mathbf{r}\text{-}\mathbf{r}')$ is the spatial delta function and $\eta(\mathbf{r})$ is the local absorption rate, with non-zero values only on the subspace of the most external electrode layers.

## Data availability

All data that support the findings of this study are available within the article and the Supplementary Information or are available from the corresponding author upon request. Source data are provided with this paper.

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

## Acknowledgements

This work was supported in part by the National Science Foundation MRSEC grant on Precision-Assembled Quantum Materials (DMR-2011738) and the National Science Foundation under grant DMR-2241180. M.S.I. was supported by a Marie Skłodowska Curie Global Fellowship (MOLCLICK: 657247) within the Horizon 2020 Programme and University of Southern California (USC) startup funds. We thank the NSF (DBI-0821671, CHE-0840366, CHE-1048807) and the NIH (S10 RR25432) for USC-based analytical instrumentation. M.C.-G., D.H.-P., and F.E. acknowledge financial support from the German Research Foundation (DFG) through Research Training Group (GRK) 1570 and Collaborative Research Center (SFB) 1277 - Project ID 314695032 (subprojects A03, B01). We thank Brandon Fowler and Nils Rotthowe for help with Mass-Spectroscopy, Giacomo Lovat for help with STM-BJ data acquisition and Rachel Austin for discussions.

## Author contributions

W.L., M.S.I., and L.V. conceived the idea and designed this work. W.L. carried out all experimental measurements and analyzed the data. M.S.I. synthesized and characterized the molecules **1**, **2**, and **3**. L.L. performed the DFT calculations with help from M.C.-G., D.H.-P., and F.E. L.V., M.S.I., F.E., and X.R. supervised the research. The manuscript was written by W.L. and L.V. with contributions from all other authors.

## Competing interests

The authors declare no competing interests.
