## [Peer Review File · Nature Communications]

Photooxidation driven formation of Fe-Au linked ferrocene-based single-molecule junctionsREVIEWER COMMENTS

Reviewer #1 (Remarks to the Author):

Lee et al. focused on the generation of metal-metal contacts by applying a photoredox reaction in single-molecule junctions. They claimed to observe direct bond formation between a ferrocene iron center and the Au electrode and reveal the role of Fe-Au bond on the conductance of single-molecule devices. Overall, the manuscript is well organized and the experimental results are well presented. My major concern raised to against their main conclusion that the observation of Fe³⁺-Au bond with gold tip in contact of ferrocene (Fc) units. First of all, the Fc-Au interaction is very weak and the Fc is almost the most stable organometallic compound. There is only one old publication from 1974 saying the Fe from Fc can be bonded with HgCl₂ ([https://doi.org/10.1016/S0022-328X\(00\)84841-X](https://doi.org/10.1016/S0022-328X(00)84841-X)) This means the direct probing Au-Fc interaction unlikely happen. Second, The Fc unit is the standard electrochemistry probe, thanks to the very strong interaction between Fc⁺ to the counterions, here the PF₆⁻. Given this strong pairing of Fc⁺ with PF₆⁻, any additional bond formation, such as Fe-Au, in the absence of extra ligands, would likely lead to the immediate decomposition of Fc. Third, the DFT calculation performed in the study appears to be somewhat superficial, as it solely encompasses the calculation of adsorption energy. Such data alone may not suffice to assert the formation of a bond. It would be beneficial if the authors could provide further insight, perhaps by identifying the molecular orbitals and d electrons of Fe involved in the formation of the Fe-Au bond. The d orbitals from Fe are all hybridized with Cp ring, where is the extra bonding possibility? To substantiate the rather unusual claim of Fe-Au bond formation, the authors need to give more evidences of chemistry proof (spectra, crystal structure, etc). Those proof can be obtained ex-situ, if they can be logically linked to the single molecular data. Unfortunately, based on the current data, I cannot agree with author's conclusion and therefore cannot support the publication of this manuscript.

In addition, some questions/comments need to be addressed as well:

1. The authors draw very small gold tips and this is very misleading to readers. The STM tips can be seen by naked eyes! Even the pointy part of the STM tip should be much larger than a ferrocene unit. The schematic illustration should be improved to a more rigorous level.
2. Can the authors measure the conductance of [1]⁺, where the tip is directly connected to SME instead of Fc, to analyze the differences for the red curves in panel b of Fig.2, which helps to make more clear how the formation of Fe-Au bond affects the conductance of the molecules.
3. Why the conductance of 2H was nearly half of the 1H giving that the length of the two molecular junctions were similar.
4. In page 7 line 133, the authors proposed that "2 forms a molecular junction in only its oxidized state". In Fig. 3a, if the Au-Fe bond was formed by applying 450 mV bias, and Fc became its neutral state at 200 ms with a bias of 100 mV. What is the configuration of the molecular junction at 200 ms with a bias of 100 mV? The Au-Fe bond was broken and the Au-SMe bond was re-formed?
5. In Fig. 5, the transmission function of 1H is missing.
6. In Fig. S2, why the displacement of 3L under the bias of 450 mV and 100 mV differed distinctively, while the contacts for them were the same.
7. There are typos should be carefully checked and revised, for example, in ref 10, it should not be 105; ref 30, it should be Agrait not Agrait.

Reviewer #2 (Remarks to the Author):

This work by Lee et al. aims to expand the toolkit of available surface chemistries for the formation of single-molecule junctions. Specifically, the authors employ a photoredox reaction to bind (oxidized) Fe centers in ferrocene directly to Au substrates. They perform a range of control experiments, including with different ferrocene derivatives and under different experimental conditions (w/wo illumination and chemical auxiliary), and complement their study with DFT-based electronic structure calculations, structural characterisation and solution electrochemistry. Single-molecule STM break junction experiments at different bias voltages are key to support their

hypothesis, namely that a direct Au/Fe bond forms during oxidative addition, and these results are further supported by break-off distance data and noise measurements. While collectively the evidence provided largely follows the authors' expectation, surface spectroscopic data or electrochemical characterisation of ferrocene bound to the Au substrate (thin-film voltammetry) would provide further direct evidence for the formation of the postulated Fe(III)-Au bond. Given that the authors already include solution-based voltammetry data, the latter should be relatively straightforward to obtain (as would XPS data, in fact). Ultimately, electrochemical STM would be a desirable tool for the present study, as it would allow for more precise control of the redox state of the system and more comparable bias measurements (as the bias can be kept constant while changing the gate voltage). That would also allow for more detailed (higher resolution) mapping of the conductance-electrochemical potential characteristics and it is interesting to speculate whether some Nernstian-type behavior may be observed in the results. Hence, in the present study the number data points (bias) is relatively small and while the authors' interpretation seems to be in line with the results, changing the medium (solvent) and the bias can have other unexpected effects. Hence, the manuscript does feel somewhat unfinished in that arguments are not fully developed (e.g., p. 3 (bottom): "By turning on or off the laser, we can access a ferrocene-based single-molecule device in which the interfacial contact (and junction conductance) varies with the charge state of the molecule." (by that point in the manuscript, I do not think that is obvious and the authors should elaborate this point further; p. 5: "We note that the measured plateau length...by 5-8 Å...as they relax and reorganize." (possibly, but snap-back can be determined and the values stated come somewhat out of the blue); p. 7: "These experimental results further the existence of the Fe-Au bond..." (again, please explain how those results support the existence of the Fe-Au bond rather than just existence of another conductance state). Hence, the manuscript requires substantial work to make the arguments clearer and more accessible to the reader. In this context, the authors should add meaningful errors or an appreciation of the uncertainty to all results, including the measured conductance, break-off distance and also the noise measurements (on a related note, the authors find an exponent of 1.3, rather than 1 or 2 for the two charge transport mechanisms stated, but do not really discuss the deviation from 1, if indeed charge transport is "through bond"; why did they only use the frequency range from 100-1000 Hz for their fit?).

Finally, there is at least one statement, which may not be correct and I would like the authors to explain further, namely that in a unipolar solvent the redox state of a molecule can not be changed due to changes in bias (p. 5). In the total absence of ions at the interface (which may be experimentally unrealistic, even in unipolar solvents), one would expect the potential drop between substrate and tip to be linear. If the potential at the redox site were to remain unchanged upon a change in applied bias, this would require the potential drop in the junction to be symmetric with regards to the redox site. In reality, however, the junction is rather asymmetric, considering the different electrode sizes, surface properties, molecule/electrode couplings etc. The authors should provide evidence to support their statement. The situation would of course be different in an electrochemical (three-electrode) setup, because in the absence of ions the potential drop at the working electrode and hence the electric field would be extremely low - but that is very different from the situation in an STM junction, in my opinion.

So overall I think the manuscript contains some very novel and interesting work, as the kind of ferrocene-Au chemistry is new to single-molecule electronics. In that sense, the work is rather specialist, even though the authors point to other areas of application (albeit not demonstrated or elaborated on, see abstract). The manuscript does need significant work, as outlined above, before it can be considered for publication.

Reviewer #3 (Remarks to the Author):

In this paper the question of local chemical bond formation by oxidation of an organic molecule, derivatives of ferrocene in this case, is addressed. There are numerous examples in the literature demonstrating the change of reactivity in excited molecular states, i.e., this question per se is not new. However, the detection of this change of functionality on an atomistic scale by (presumably) removing one electron from the HOMO is demonstrated quite convincingly in this paper, which represents clearly a significant progress.

A series of control experiments based on the scanning tunneling microscope-based break junction (STM-BJ) technique were carried out and are complemented by calculations of transmission probabilities based on density functional theory. By changing functional end groups and varying the oxidative properties of the environment, by optical excitation and variation of distance between contacts, together with noise analysis, the authors collect signs of evidence that a direct bond between the central Fe ion and a gold tip is established for the molecule being in the oxidized state. No such bond is formed for the neutral molecule. Here bonds are only formed through the end groups. The main indicators for the formation of a direct bond between Au and Fe³⁺ are a significant change of conductance through the molecule (at least a factor of 3 under directly comparable conditions) and the difference in length of conductance plateaus.

Therefore, the paper deserves publication at some stage.

However, the paper as it stands is more written for the specialist than for a general audience. It also does not convey a clear message given the fact that the methods and techniques used here far from „applications“ or the use in „devices“. Generally, bond formation of initially inert materials in an excited state is well known.

Furthermore, the paper contains many quite complicated and intriguing aspects whose essentials are partly distributed between the main paper and the supplementary (e.g. Fig. S2) so that it is impossible to understand the contents of the main paper without studying in detail the supplementary.

Details:

Photoredox measurements: Are you assuming that basically 100% of the ferrocene derivatives are in their excited state and survive in this state during conductance measurements?

Are there any fluctuations (due to limited lifetime) observable?

What is the probability to find a molecule in its excited state? Was the laser intensity varied, and what is the result? There must be also a possibility for de-excitation within the junction, i.e. in contact with the Au electrodes at low voltages.

Fig. 1: From this figure, one gets the (false) impression that two tips are involved instead of a tip and a surface.

Fig. S4: Why is the conductance peak shifting back after 10 min?

Fig.2, caption: „3 forms only “low-conducting” junction geometries in either oxidized or neutral states.“ This statement is obviously wrong, since comparing plot 1 and 2 with 3 the plotted conduction is lowest at the highest voltage, i.e. it is reversed compared to 1 and 2. Similarly, lines 140 to 142 should be modified.

I. 223: This sounds like auto-ionization. How realistic is this description? In other words, what would happen if the Au clusters are connected with macroscopic contacts?

REVIEWER COMMENTS

Reviewer #1 (Remarks to the Author):

Comments:

Lee et al. focused on the generation of metal-metal contacts by applying a photoredox reaction in single-molecule junctions. They claimed to observe direct bond formation between a ferrocene iron center and the Au electrode and reveal the role of Fe-Au bond on the conductance of single-molecule devices. Overall, the manuscript is well organized and the experimental results are well presented. My major concern raised to against their main conclusion that the observation of Fe³⁺-Au bond with gold tip in contact of ferrocene (Fc) units.

Response: We thank the reviewer for carefully reading our manuscript and providing constructive comments. We have addressed them all in our revised manuscript as detailed below.

First of all, the Fc-Au interaction is very weak and the Fc is almost the most stable organometallic compound. There is only one old publication from 1974 saying the Fe from Fc can be bonded with HgCl₂ ([https://doi.org/10.1016/S0022-328X\(00\)84841-X](https://doi.org/10.1016/S0022-328X(00)84841-X)) This means the direct probing Au-Fc interaction unlikely happen. line

Response: Thank you for bringing up this point. Yes – the Fc-Au interaction is not very strong: the calculated binding energy is ~ 0.80 eV as shown in our calculation (**Fig. S10**). However, many other linkers we have used, such as amine, methyl-sulfide, phosphine, pyridine, etc. have comparable binding energies ranging from 0.5 to 1.2 eV.¹⁻⁸ They bind weakly to uncoordinated gold, allowing us to measure the single-molecule conductance of over many many different molecular backbones.⁹⁻¹⁸

We would like to highlight the method of our measurement in STM-BJ measurements, which starts with the formation of a gold-gold contact that is thinned out exposing undercoordinated and highly reactive gold – it is these gold atoms on the surface of the electrode that bind to the ferrocene iron center. We do not believe that oxidized ferrocene will bind to a Au(111) surface through the formation of an Fe-Au bond since the surface gold atoms are not sufficiently undercoordinated.

We stress further that direct metal bonds to a ferrocene center with other metals have been observed after 1974, though in the presence of supporting ligands as referred to in our revised manuscript (references 16-23). In our work, taking advantage of our STM-BJ setup's capability to measure the conductance of single-molecule junction and using the electrochemical behaviors of ferrocene derivatives, we form the Fe-Au bonds to undercoordinated gold, without the need for synthetic ligand-based stabilization.

Second, The Fc unit is the standard electrochemistry probe, thanks to the very strong interaction between Fc⁺ to the counterions, here the PF₆⁻. Given this strong pairing of Fc⁺ with PF₆⁻, any additional bond formation, such as Fe-Au, in the absence of extra ligands, would likely lead to the immediate decomposition of Fc.

Response: This is an interesting point which lead us to some additional experiments. In our original manuscript, the anion, [PF₆]⁻, was used involved only in the light measurements to induce the photoredox reaction as we showed in **Fig. 1**. All other STM-BJ measurements were made without any [PF₆]⁻ species. In response to the reviewer's concern, we show below STM-BJ

measurement of molecule **1** with ~ 0.1 equiv. of $[\text{R}_2\text{I}]^+[\text{PF}_6]^-$, keeping the same condition as the photoredox measurements (**Fig. R1a**). The results are the same as the measurement without the anion in **Fig. 2a**, implying that the $[\text{PF}_6]^-$ does not affect the junction formations proposed here. Furthermore, we have measured the conductance of **1** with an excess of $[\text{PF}_6]^-$ (13 equiv. TBAPF₆) to confirm that it doesn't affect the measurements. The results are shown in **Fig. R1b**. A high concentration of iodine hinders the formation of molecular junctions, as evidenced by a decrease in the height of the $1 G_0$ peak, corresponding to the formation of single Au-Au atomic point contacts (**Fig. R1**). Thus, we used TBAPF₆ to examine the effect of a high concentration of $[\text{PF}_6]^-$ on junction formation. Again, the results are the same as those made without the counterion. Therefore, we can exclude the possibility that we cannot form a junction in the presence of $[\text{PF}_6]^-$.

We would like to note that the stability of oxidized ferrocene versus other redox-active compounds is largely due to the fact you take the electron out of a non-bonding orbital, not ion pairing (an electrostatic interaction) with counterions. Ferrocenium may be readily isolated with weakly coordinating ions such as BARF⁻ (tetrakis[3,5-bis(trifluoromethyl)phenyl]borate), see for example: *Dalton Trans.*, 2021, **50**, 7433-7455. While ferrocenium is known to exhibit chemical instability in bulk solution under specific conditions, reacting with organic solvents or air, this only serves to further highlight the plausibility of molecule-electrode chemical interactions in these experiments. The transient nature of the molecular junctions formed also means that ferrocenium-electrode nanostructures may be formed and observed prior to any possible decomposition reactions, which may occur at much slower rates.

Fig. R1. STM-BJ measurements of **1** with (a) ~ 0.1 equiv. of $[\text{R}_2\text{I}]^+[\text{PF}_6]^-$ and (b) ~ 13 equiv. of TBAPF₆ in propylene carbonate (PC) at three different bias voltages.

We have added these figures to our revised SI document. (**Fig. S2**) and line 67-69 in our revised manuscript.

Third, the DFT calculation performed in the study appears to be somewhat superficial, as it solely encompasses the calculation of adsorption energy. Such data alone may not suffice to assert the formation of a bond. It would be beneficial if the authors could provide further insight, perhaps by identifying the molecular orbitals and d electrons of Fe involved in the formation of

the Fe-Au bond. The d orbitals from Fe are all hybridized with Cp ring, where is the extra bonding possibility?

Response: Thank you for bringing up this point. The molecular orbital diagram of ferrocene with D_{5h} symmetry is shown in **Fig. R2a**.^{19,20} The HOMO orbital is an Fe atom based d_{z^2} orbital. Once the ferrocene is oxidized, the singly occupied d_{z^2} orbital (SOMO) will participate in the bond formation with the Au. We note that the π -orbitals of Cp rings do not contribute the HOMO (or SOMO) d_{z^2} orbitals. Furthermore, we calculated the spin density which shows that the d_{z^2} orbital is located on the iron center (**Fig. R2b**). This is consistent with the molecular orbital diagram, suggesting that the iron center is the most favorable site for binding to the gold electrode in its oxidized state.

We elaborate on this point: the stability of the electronic structure of the frontier orbitals under molecular deformation expresses itself in the spectral gap between the e_{2g} , a_{1g} , e_{1g} orbitals. This gap is of the order of 1eV and therefore it is not small; an important stabilizing factor is the D_{5h} -symmetry, which allows for a residual degeneracy in the HOMO-1 (e_{2g}) and LUMO (e_{1g}) orbitals.

On the other hand, unsaturated Au-atoms are highly reactive because they exhibit a weakly-split multiplet of 3d-orbitals. Therefore, in the situation where a ferrocene meets an unsaturated Au-atom two energy scales compete neither of which is small: an energy cost arises for reducing the D_{5h} symmetry, i.e. mixing in d_{xz} - and d_{yz} -orbitals (e_{1g}), whereas an energy gain results from the new bond between the d-orbitals of Fe and Au. Without an explicit calculation, the outcome of this competition is difficult to predict. The result of our ab-initio study is, on the other hand, very clear and points towards the possibility of a direct binding with a binding energy in the typical range.

We hope that our discussion also answers the reviewer's question where the extra bonding-possibility comes from: it is a result of mixing in d_{xz} - and d_{yz} -orbitals.

In response to the reviewer's comment, we have added these figures and associated discussion to our revised SI document (**Fig. S16**) and line 265-267 in our revised manuscript.

Fig. R2. (a) The molecular orbital diagram of ferrocene with D_{5h} symmetry. (b) Calculated spin density distribution of the Fc-Au contact. Light blue and yellow regions denote spin-up and spin-down densities, respectively. Those antiparallel spins interact with each other, forming the Fe-Au bond. To make the spin density visible, the isosurface value is set to be 10^{-7} .

To substantiate the rather unusual claim of Fe-Au bond formation, the authors need to give more evidences of chemistry proof (spectra, crystal structure, etc). Those proof can be obtained ex-situ, if they can be logically linked to the single molecular data.

Response: Thank you for the suggestion. As outlined in the manuscript, researchers have characterized the formation of ligand-supported Fe-M bonds using X-ray crystallography, NMR, and X-ray absorption near-edge spectroscopy (XANES), etc. We acknowledge the need for follow-up studies to validate the formation of the Fe-Au bond. However, obtaining crystal structure of the molecules bound in the junction, linked by the Fe-Au bond is currently unattainable either in situ or ex situ as we discussed above since we need reactive undercoordinated Au atoms for such measurements. Additionally, we need the Fe to be in an oxidized state, which makes it incompatible with vacuum requirements of XPS or XANES. In our experiments, we introduce uncoordinated gold atoms by rupturing the Au-Au contact during STM-BJ measurements, facilitating the formation of the Fe-Au bond. Additionally, we manipulate the oxidation states of ferrocene derivatives to create the Fc-Au bond in a solution system.

Unfortunately, based on the current data, I cannot agree with author's conclusion and therefore cannot support the publication of this manuscript.

Response: We realize that our conclusion is unexpected – indeed we reached this conclusion through a process of elimination. First, we clearly see junctions being formed when the ferrocene is in the oxidized state. This is evident from the peaks seen in both the 1D and 2D conductance histograms, based on the measurements where we can hold a junction, based on the flicker noise data and based on our experience in measuring single-molecule junctions. This leaves two possible locations where the Fc unit can bind to gold – (1) through the Cp ring through van der Waals (vdW) interaction and (2) through the formation of an Fe-Au donor-acceptor interaction. Our calculations indicate that the vdW interaction between the Cp ring and Au is very weak (0.2-0.3 eV), making it very unlikely that we can sustain a junction for example for 150 ms as we show in Figure 3. Furthermore, such a vdW coupled junction would give a flicker noise signature that is different from what we observe in the experiments shown in Figure 4. Finally, if vdW interactions were sufficient to sustain a junction, we would not need to oxidize the molecule to form the junction.

Having ruled out the possibility that we are forming a junction through the Cp ring, we turn to the evidence we have supporting the formation of an Fe-Au bond. First, our calculations show that such a bond has a binding energy well in the range of other linkers we have used successfully for measurements using the STM-BJ method. Second, the flicker noise data indicates that we have a bond that does not show large conductance fluctuations initiated by thermal motion of atoms in the junction (unlike what we would expect for a vdW coupled junction). Finally, as we show above in our new calculations, when the Fc is oxidized, the d_{z^2} orbital on the Fe is available for forming a bond with the Au.

Although we agree in principle that a crystal structure of such a junction would be ideal to confirm bond formation, we believe this is really outside the scope of this work.

In addition, some questions/comments need to be addressed as well:

1. The authors draw very small gold tips and this is very misleading to readers. The STM tips can be seen by naked eyes! Even the pointy part of the STM tip should be much larger than a ferrocene unit. The schematic illustration should be improved to a more rigorous level.

Our cartoon representation of the junction focuses on the very apex of the STM tip and substrate where the molecule binds. As described in the Methods section, a piezo actuator drives a gold tip, which is pushed into contact with a gold substrate, forming a gold-gold contact. Following this, the gold tip is retracted, forming a gold point contact that is thinned down to a single-atom contact. This single-atom contact has a conductance of $1G_0$, which is why all our conductance histograms have a peak at integer multiples of $1G_0$. Once this contact is broken, a nanoscale gap is created between an atomically sharp gold tip and a protrusion on the substrate. Since we start with a gold single-atom contact it is quite likely that at the 3-4 layer length scale, we have pyramidal structures that bind to the molecule.^{21,22} This is supported by transmission electron microscope (TEM) images.²³ Furthermore, our illustration approach follows the conventional practices in the field of single-molecule junction electronics.²⁴⁻²⁶ If the reviewer has a specific suggestion on how our cartoon should be improved, we would be happy to follow their suggestion.

2. Can the authors measure the conductance of $[1]^+$, where the tip is directly connected to SMe instead of Fc, to analyze the differences for the red curves in panel b of Fig.2, which helps to make more clear how the formation of Fe-Au bond affects the conductance of the molecules.

Response: Thank you for the suggestion. We measured the conductance of $[1]^+$ with the **1L** geometry, where only -SMe linkers are involved in the junction formation, by utilizing the hold measurement. First, we formed the **1H** geometry at a starting bias of 450 mV and maintain the voltage while holding the junction. The conductance is stable during the hold and around $10^{-3} G_0$ (**Fig. R3a**). We repeat this measurement starting at a lower bias of 200 mV, the junction starts in the **1L** geometry as the bias is too low to oxidize the ferrocene. If we increased the voltage to 450 mV while holding the molecule in the **1L** geometry, we observe a different conductance. In principle, this bias should oxidize **1** to the $[1]^+$ state while the molecule is bound to the electrodes through the terminal -SMe linkers. The conductance when the bias is increased shows a slight increase but does not reach $10^{-3} G_0$ (**Fig. R3b**). Given that the conductance values of **1H** and **1L** are significantly distinct from each other, it clearly shows that the different junction geometries have different conductances in the oxidized state as well. We have included these results in the SI (**Fig. S7**) and line 166-167 in our revised manuscript.

Fig. R3. STM-BJ hold measurements. Top: The switching bias ramp applied across molecular junctions as a function of time. Bottom: 2D conductance-time histograms obtained for **1** using different bias ramps. (a) Starting at a bias of 450 mV to form the **1H** junction and then maintaining the bias at 450 mV. (b) Starting with a bias of 200 mV to form the **1L** junction and then changing the bias to 450 mV to oxidize it. Note that the conductance during the central portion of the hold segment is different depending on the bias at the start of the measurement as shown by the dashed lines clearly indicating that the **1H** and **1L** geometries are distinct junctions.

3. Why the conductance of 2H was nearly half of the 1H giving that the length of the two molecular junctions were similar.

Response: Thank you for asking this question. Despite the similarity in the electron pathway of **1H** and **2H**, the presence of an additional thioanisole group in **1** likely alters the direction of an electrode linked to Fe, resulting in a conductance difference between **1H** and **2H**. To confirm this, we have calculated the transmission of **1H** in our revision and find that it is approximately a factor of two higher than that of **2H** similar to what we observe in the experiments. We have added this new calculation and additional details to the SI (**Fig. S14**) and line 232-234 / 257-260 in our revised manuscript.

Fig. R4. (a) The relaxed junction geometries for (a) **1H** and (b) **2H**. Dark grey, light grey, red, yellow, and gold spheres represent C, H, Fe, S, Au atoms, respectively. (c) Calculated transmission functions for **1H** and **2H**.

4. In page 7 line 133, the authors proposed that “2 forms a molecular junction in only its oxidized state”. In Fig. 3a, if the Au-Fe bond was formed by applying 450 mV bias, and Fc became its neutral state at 200 ms with a bias of 100 mV. What is the configuration of the molecular junction at 200 ms with a bias of 100 mV? The Au-Fe bond was broken and the Au-SMe bond was re-formed?

Response: The hold measurements depicted in **Fig. 3** are conducted with **1**, not **2**. The **1H** geometry represents a junction linked by the SMe linker on one side and the ferrocene iron center on the other while the **1L** geometry represents a junction linked by SMe linkers on both sides as depicted in **Fig. 2a**. The configuration of the molecular junction at 200 ms with a bias of 100 mV is the **1L** geometry, as shown in **Fig. 3b**.

5. In Fig. 5, the transmission function of 1H is missing.

Response: Thank you for bringing up this. We have calculated the transmission of **1H** and added it to the SI in our revision (**Fig. S14**).

6. In Fig. S2, why the displacement of 3L under the bias of 450 mV and 100 mV differed distinctively, while the contacts for them were the same.

Response: As we mentioned in the caption of **Fig. S3**, the length of LUMO-conducting molecules shortens as the applied bias increases in polar solvents, which is aligned with our previous observations (Reference 10 in SI). Given that **1** and **2** exhibit higher conductance at more positive bias voltages, those are HOMO-conducting molecules. In contrast, **3** exhibits higher conductance at more negative bias voltages, indicating it is a LUMO-conducting molecule. To clarify this point, we have added explanations in the caption of **Fig. S3**. The

underlying reason for these different lengths is not understood but seen with many different molecules.

7. There are typos should be carefully checked and revised, for example, in ref 10, it should not be 105; ref 30, it should be Agraït not Agrait.

Response: Thank you for pointing this out. We have corrected the typos.

Reviewer #2 (Remarks to the Author):

This work by Lee et al. aims to expand the toolkit of available surface chemistries for the formation of single-molecule junctions. Specifically, the authors employ a photoredox reaction to bind (oxidized) Fe centers in ferrocene directly to Au substrates. They perform a range of control experiments, including with different ferrocene derivatives and under different experimental conditions (w/wo illumination and chemical auxiliary), and complement their study with DFT-based electronic structure calculations, structural characterisation and solution electrochemistry. Single-molecule STM break junction experiments at different bias voltages are key to support their hypothesis, namely that a direct Au/Fe bond forms during oxidative addition, and these results are further supported by break-off distance data and noise measurements. While collectively the evidence provided largely follows the authors' expectation, surface spectroscopic data or electrochemical characterisation of ferrocene bound to the Au substrate (thin-film voltammetry) would provide further direct evidence for the formation of the postulated Fe(III)-Au bond.

Response: We thank the reviewer for carefully reading our manuscript and providing constructive comments. We have addressed them all in our revised manuscript as detailed below.

Given that that the authors already include solution-based voltammetry data, the latter should be relatively straightforward to obtain (as would XPS data, in fact).

Response: Thank you for the suggestion. As outlined in the manuscript, researchers have characterized the formation of ligand-supported Fe-M bonds using X-ray crystallography, NMR, and X-ray absorption near-edge spectroscopy (XANES), etc. We acknowledge the need for follow-up studies to validate the formation of the Fe-Au bond. However, it's important to note that obtaining crystal structure spectra of the molecules bound in the junction, linked by the Fe-Au bond, is currently unattainable either in situ or ex situ. In our approach, we introduce uncoordinated gold atoms by rupturing the Au-Au contact during STM-BJ measurements, facilitating the formation of the Fe-Au bond. Additionally, we manipulate the oxidation states of ferrocene derivatives to create the Fc-Au bond in the solution system. We anticipate that the resulting bond has a relatively short life time (~ few minutes) at room temperature given the binding energy of 0.8 eV, preventing the preparation of stable compounds bearing the Fe-Au bond for ex-situ characterizations. Furthermore, XPS or other such surface-science characterizations are typically made on gold that does not have undercoordinated atoms (i.e. Au(111)) and in ultrahigh vacuum. This makes it difficult to investigate a bond that occurs between the oxidized Fc and undercoordinated Au.

Ultimately, electrochemical STM would be a desirable tool for the present study, as it would allow for more precise control of the redox state of the system and more comparable bias measurements (as the bias can be kept constant while changing the gate voltage). That would also allow for more detailed (higher resolution) mapping of the conductance-electrochemical potential characteristics and it is interesting to speculate whether some Nernstian-type behavior may be observed in the results. Hence, in the present study the number data points (bias) is relatively small and while the authors' interpretation seems to be in line with the results, changing the medium (solvent) and the bias can have other unexpected effects.

Response: Thank you for the suggestions. Yes – electrochemical STM (three-electrode system) would be a good tool for studying electrochemical characteristics of ferrocene derivatives. In our

STM-BJ experimental setup we use a two-electrode system where the tip serves as a microelectrode and working electrode while the substrate is counter and reference.²⁷ The charge state of molecules around the tip electrode is changed by the tip bias. We and others have shown that this setup is the same as using a third gate electrode.²⁸⁻³⁰ Additionally, we do not agree that utilizing a three-electrode system provides more precise control of the redox state and enables mapping at a higher resolution of the conductance-potential data. While applying potential with a gate electrode, we still need to measure a few thousand traces (~ a few hours) to determine the average conductance at each potential voltage. In other words, introducing a three-electrode system (gate electrode) does not necessarily offer an advantage in mapping continuous data points. We note that gating experiments cannot be performed on individual junctions (“hold” measurements) as it takes time for the electrolytes to reorganize around the junction.

In response to the reviewer’s concern, we have included additional bias voltage data points as shown in **Fig. R5**. The conductance peak shifts as the bias voltage changes. Moreover, we observe a Nernstian-type behavior in ferrocene derivatives. At around the redox potential of **1** (~ 300 mV), the broad conductance peaks suggest that both **1H** and **1L** are formed due to the coexistence of oxidized and neutral species.

Fig. R5. (a) 1D conductance histogram of **1** in PC at the range between 550 mV to -750 mV. (b) Average conductance at each bias voltage, fitted using Gaussian functions.

Furthermore, as we already discussed in the manuscript, changing polarity of solvents affects the formation of the Fe-Au bond during the STM-BJ measurements. To support this, we have

included an additional experiment demonstrating that **2** does not form a junction even at high bias voltages in tetradecane, a highly non-polar solvent (**Fig. R6**). The ferrocene derivatives are not oxidized in tetradecane with the applied tip bias voltage.

Fig. R6. (a) 1D conductance histograms of **2** measured at 100 mV (light violet), 450 mV (light pink), and 750 mV (pink) in a tetradecane, a non-polar solvent. (b, c) 2D conductance-displacement histograms of **2** at 450 and 750 mV.

Hence, the manuscript does feel somewhat unfinished in that arguments are not fully developed (e.g., p. 3 (bottom): "By turning on or off the laser, we can access a ferrocene-based single-molecule device in which the interfacial contact (and junction conductance) varies with the charge state of the molecule." (by that point in the manuscript, I do not think that is obvious and the authors should elaborate this point further; p. 5: "We note that the measured plateau length...by 5-8 Å...as they relax and reorganize." (possibly, but snap-back can be determined and the values stated come somewhat out of the blue); p. 7: "These experimental results further the existence of the Fe-Au bond..." (again, please explain how those results support the existence of the Fe-Au bond rather than just existence of another conductance state).

Response: Thank you for the constructive comments. We have carefully revised these paragraphs (line 72-74) to enhance clarity. Specifically, in addressing the snapback distance, we have provided a more explicit explanation and included an additional reference in the main manuscript (line 95-98 and 151-154). [Quek, S. Y. et al. Nature nanotechnology 4, 230-234 (2009)] Furthermore, we have included another reference that reports the snapback distance in propylene carbonate solvent. [Zhang, M. et al. JACS 145, 6480-6485 (2023)]

Hence, the manuscript requires substantial work to make the arguments clearer and more accessible to the reader. In this context, the authors should add meaningful errors or an appreciation of the uncertainty to all results, including the measured conductance, break-off distance and also the noise measurements (on a related note, the authors find an exponent of 1.3, rather than 1 or 2 for the two charge transport mechanisms stated, but do not really discuss the deviation from 1, if indeed charge transport is "through bond"; why did they only use the frequency range from 100-1000 Hz for their fit?).

Response: Thank you for the constructive comments. We added experimental errors for conductance measurements in the Methods section (line 304-305). The error for the break-off distance is not necessary because molecular junctions are formed right after the gold point contact is broken so the starting distance is the same for all traces in the 2D histogram. As shown in 2D histograms, the breakpoint is aligned to zero displacement for every measured trace.

As previously reported, the rotation of Cp rings introduces quantum interference in electron transport through the junction.³¹ The deviation in the exponent can be attributed to this quantum interference from the rotation of Cp rings, and this point has been added to the revision in the main manuscript (line 194-196). The deviation in the power-law dependence has been consistently observed across various molecules.³²⁻³⁵ Additionally, as we explained in the Methods section, we use a range of 100 to 1000 Hz for the flicker noise analysis as below 100 Hz, we have noise due to the power line (60 Hz) and some mechanical vibrations. We avoid the region above 1000 Hz to avoid any noise introduced by our current amplifier at the 100 kHz acquisition rate.

Finally, there is at least one statement, which may not be correct and I would like the authors to explain further, namely that in a unpolar solvent the redox state of a molecule can not be changed due to changes in bias (p. 5). In the total absence of ions at the interface (which may be experimentally unrealistic, even in unpolar solvents), one would expect the potential drop between substrate and tip to be linear. If the potential at the redox site were to remain unchanged upon a change in applied bias, this would require the potential drop in the junction to be symmetric with regards to the redox site. In reality, however, the junction is rather asymmetric, considering the different electrode sizes, surface properties, molecule/electrode couplings etc. The authors should provide evidence to support their statement. The situation would of course be different in an electrochemical (three-electrode) setup, because in the absence of ions the potential drop at the working electrode and hence the electric field would be extremely low - but that is very different from the situation in an STM junction, in my opinion.

Response: Thank you for bringing up this point. As we mentioned in the Methods section, we prepared a wax-coated tip to reduce the exposed surface. Such a coated tip with a very small area builds up a dense double layer around it in polar solvents allowing us to oxidize or reduce the molecule in the junction using a positive or negative tip bias.^{36,37} In a non-polar solvent, we cannot build up a dense double layer around the uncoated tip because these solvents do not support a sufficient ion concentration. As a result, most molecules around the tip remain in a neutral charge state even at high bias voltages. To support this, we show below a cyclic voltammetry (CV) measurement of **1** (**Fig. R8**) in a non-polar solvent. As can be seen, we do not measure any current when sweeping the voltage due to Faradaic processes. This further explains why we are not able to form a junction of **2** in a non-polar solvent even at high bias voltages as shown in **Fig. R6**.

Fig. R8. *In situ* cyclic voltammetry (CV) measurements of **1** in a non-polar solvent, tetradecane (TD).

So overall I think the manuscript contains some very novel and interesting work, as the kind of ferrocene-Au chemistry is new to single-molecule electronics. In that sense, the work is rather specialist, even though the authors point to other areas of application (albeit not demonstrated or elaborated on, see abstract). The manuscript does need significant work, as outlined above, before it can be considered for publication.

Response: Thank you for these constructive comments.

Reviewer #3 (Remarks to the Author):

In this paper the question of local chemical bond formation by oxidation of an organic molecule, derivatives of ferrocene in this case, is addressed. There are numerous examples in the literature demonstrating the change of reactivity in excited molecular states, i.e., this question per se is not new. However, the detection of this change of functionality on an atomistic scale by (presumably) removing one electron from the HOMO is demonstrated quite convincingly in this paper, which represents clearly a significant progress.

A series of control experiments based on the scanning tunneling microscope-based break junction (STM-BJ) technique were carried out and are complemented by calculations of transmission probabilities based on density functional theory. By changing functional end groups and varying the oxidative properties of the environment, by optical excitation and variation of distance between contacts, together with noise analysis, the authors collect signs of evidence that a direct bond between the central Fe ion and a gold tip is established for the molecule being in the oxidized state. No such bond is formed for the neutral molecule. Here bonds are only formed through the end groups. The main indicators for the formation of a direct bond between Au and Fe³⁺ are a significant change of conductance through the molecule (at least a factor of 3 under directly comparable conditions) and the difference in length of conductance plateaus.

Response: We thank the reviewer for carefully reading our manuscript and providing constructive comments. We have addressed them all in our revised manuscript as detailed below.

Therefore, the paper deserves publication at some stage.

However, the paper as it stands is more written for the specialist than for a general audience. It also does not convey a clear message given the fact that the methods and techniques used here far from „applications“ or the use in „devices“. Generally, bond formation of initially inert materials in an excited state is well known.

Furthermore, the paper contains many quite complicated and intriguing aspects whose essentials are partly distributed between the main paper and the supplementary (e.g. Fig. S2) so that it is impossible to understand the contents of the main paper without studying in detail the supplementary.

Response: Thank you for the constructive comments. As will be discussed later, we have addressed all the comments. This work is novel as it reports, for the first time, the formation of a Fe-Au bond without synthetic support. Our findings mark the initial step toward the development of a new generation of electrical devices that aim for enhanced efficiency through reduced size and increased conductivity. Moreover, numerous fundamental discoveries using the STM-BJ technique have recently been published in Nature Communications.^{9,28,32,38-40} We are thus a bit perplexed by the comments relating this work to a specialized audience. We hope that our revision will provide sufficient additional details to enable a general audience to understand our work.

Details:

Photoredox measurements: Are you assuming that basically 100% of the ferrocene derivatives are in their excited state and survive in this state during conductance measurements?

Response: Thank you for asking this question. We note that light is used to excite **1** (one of the ferrocene derivatives in question), whereby the molecule in its excited state is then converted to

the oxidized form through reaction with R_2I^+ (Fig 1(a)). It is the oxidized form of the ferrocene that is measured during conductance measurements, not the excited state. The reviewer might be referring to the "excited state" as the oxidized state of **1**. Certainly, when we use light to oxidize ferrocene, we see clearly that even after turning the light off, we see some traces formed with the oxidized species at low bias therefore a large fraction of the molecules must be in the oxidized state. Since we irradiated with a very high intensity of light, the prevalent species are the oxidized species. When we measure **1H** using a high bias, the population of the oxidized species follows the Nernst equation.

Are there any fluctuations (due to limited lifetime) observable?

What is the probability to find a molecule in its excited state? Was the laser intensity varied, and what is the result? There must be also a possibility for de-excitation within the junction, i.e. in contact with the Au electrodes at low voltages.

Response: Because the binding energies are sufficiently high (~ 0.8 eV), we do not observe the molecule binding and unbinding at room temperature within our measurement time scales (0.5 s).

We believe there is some misunderstanding here. Molecule **1** absorbs light and in the presence of $[R_2I]^+[PF_6]^-$ gets oxidized relatively quickly. We do not measure the molecule in its excited state because the lifetime of the excited state is short. Rather we measure the molecule once it is oxidized after it interacts with $[R_2I]^+[PF_6]^-$. Given our postulation that only the $[1]^+$ state forms the Fe-Au bond, we can conclude that the **1H** junction is formed only in its oxidized ground states.

We note that the laser intensity needs to be high (at least ~ 100 mW/cm²) to induce the photoredox reactions as previous studies suggest.^{41,42} When we irradiated the laser with lower intensities, the high-conducting junction was not formed.

Fig. 1: From this figure, one gets the (false) impression that two tips are involved instead of a tip and a surface.

Response: Thank you for pointing this out. As described in the Methods section, a piezo actuator drives a gold tip, which is pushed into contact with a gold substrate, forming a gold-gold contact. Following this, the gold tip is retracted, forming a gold point contact that is thinned down to a single-atom contact. This single-atom contact has a conductance of $1G_0$, which is why all our conductance histograms have a peak at $1G_0$. Once this contact is broken, a nanoscale gap is created between an atomically sharp gold tip and a sharp protrusion on the substrate (the latter has been imaged by STM as shown in our earlier work⁴³). Since we start with a gold single-atom contact it is quite likely that at the 3-4 layer length scale, we have pyramidal structures that bind to the molecule both on the tip and substrate side.^{21,22} Furthermore, our illustration approach follows the conventional practices in the field of single-molecule junction electronics.²⁴⁻²⁶ If the reviewer has a specific suggestion on how our cartoon should be improved, we would be happy to follow their suggestion.

Fig. S4: Why is the conductance peak shifting back after 10 min?

Response: Thank you for asking this question. The introduction of a strong oxidant leads to the formation of oxidized species, $[1]^+$, enabling the formation of the **1H** junction. However, since we apply bias voltage below the redox potential, the $[1]^+$ species around the tip are reduced back to the neutral state over time. Consequently, after 10 minutes, only the **1L** junction is formed. This observation aligns with the bias dependence STM-BJ results, as illustrated in **Fig. 2**.

Fig.2, caption: „3 forms only “low-conducting” junction geometries in either oxidized or neutral states.“ This statement is obviously wrong, since comparing plot 1 and 2 with 3 the plotted conduction is lowest at the highest voltage, i.e. it is reversed compared to 1 and 2. Similarly, lines 140 to 142 should be modified.

Response: Thank you for pointing this out. There might be a misunderstanding regarding the term "low-conducting" junction geometry. As clarified in the manuscript, it refers to a junction linked only through the MeS-Au bonds on both sides, not an Fe-Au bond on one side and a MeS-Au bond on the other. Due to the steric bulk of methyl groups in the octa-methyl substituted derivative, the Fe-Au bond formation is inhibited for **3**. Your observation of the reversed conductance trend strongly supports that the "high-conducting" junction geometry through the Fe-Au bond is not formed. As detailed in the manuscript, if the Fe-Au bond were formed above the redox potential, we would expect higher conductance.

l. 223: This sounds like auto-ionization. How realistic is this description? In other words, what would happen if the Au clusters are connected with macroscopic contacts?

Response: Thank you for asking this question. We do not believe this is an auto-ionization process, which typically involves two identical molecules. First, in our calculations, we did not include two identical molecules. Furthermore, this calculation procedure is designed to set the net charge to simulate the oxidized state while forming the molecular junctions, making it reflective of the experimental circumstances. We are uncertain about the meaning of “Au clusters are connected with macroscopic contacts”.

References

- 1 Li, H. *et al.* Electric field breakdown in single molecule junctions. *Journal of the American Chemical Society* **137**, 5028-5033 (2015).
- 2 Aradhya, S. V., Frei, M., Hybertsen, M. S. & Venkataraman, L. Van der Waals interactions at metal/organic interfaces at the single-molecule level. *Nature materials* **11**, 872-876 (2012).
- 3 Fatemi, V., Kamenetska, M., Neaton, J. & Venkataraman, L. Environmental control of single-molecule junction transport. *Nano letters* **11**, 1988-1992 (2011).
- 4 Hybertsen, M. S. *et al.* Amine-linked single-molecule circuits: systematic trends across molecular families. *Journal of physics: Condensed matter* **20**, 374115 (2008).
- 5 Kamenetska, M. *et al.* Formation and evolution of single-molecule junctions. *Physical review letters* **102**, 126803 (2009).
- 6 Parameswaran, R. *et al.* Reliable formation of single molecule junctions with air-stable diphenylphosphine linkers. *The Journal of Physical Chemistry Letters* **1**, 2114-2119 (2010).
- 7 Park, Y. S. *et al.* Contact chemistry and single-molecule conductance: a comparison of phosphines, methyl sulfides, and amines. *Journal of the American Chemical Society* **129**, 15768-15769 (2007).
- 8 Schneebeli, S. T. *et al.* Single-molecule conductance through multiple π - π -stacked benzene rings determined with direct electrode-to-benzene ring connections. *Journal of the American Chemical Society* **133**, 2136-2139 (2011).
- 9 Zang, Y. *et al.* Directing isomerization reactions of cumulenes with electric fields. *Nature Communications* **10**, 4482 (2019).
- 10 Zang, Y. *et al.* In Situ Coupling of Single Molecules Driven by Gold-Catalyzed Electrooxidation. *Angewandte Chemie International Edition* **58**, 16008-16012 (2019).
- 11 Zang, Y. *et al.* Voltage-induced single-molecule junction planarization. *Nano Letters* **21**, 673-679 (2020).
- 12 Venkataraman, L., Klare, J. E., Nuckolls, C., Hybertsen, M. S. & Steigerwald, M. L. Dependence of single-molecule junction conductance on molecular conformation. *Nature* **442**, 904-907 (2006).
- 13 Vazquez, H. *et al.* Probing the conductance superposition law in single-molecule circuits with parallel paths. *Nature Nanotechnology* **7**, 663-667 (2012).
- 14 Quek, S. Y. *et al.* Mechanically controlled binary conductance switching of a single-molecule junction. *Nature nanotechnology* **4**, 230-234 (2009).
- 15 Li, L. *et al.* Highly conducting single-molecule topological insulators based on mono- and di-radical cations. *Nature Chemistry* **14**, 1061-1067 (2022).
- 16 Lee, W. *et al.* Increased molecular conductance in oligo [n] phenylene wires by thermally enhanced dihedral planarization. *Nano Letters* **22**, 4919-4924 (2022).
- 17 Greenwald, J. E. *et al.* Highly nonlinear transport across single-molecule junctions via destructive quantum interference. *Nature Nanotechnology* **16**, 313-317 (2021).
- 18 Cheng, Z.-L. *et al.* In situ formation of highly conducting covalent Au-C contacts for single-molecule junctions. *Nature nanotechnology* **6**, 353-357 (2011).
- 19 Rahnamaye Aliabad, H. & Chahkandi, M. Comprehensive SPHYB and B3LYP-DFT Studies of Two Types of Ferrocene. *Zeitschrift für anorganische und allgemeine Chemie* **643**, 420-431 (2017).

- 20 Yamaguchi, Y. & Kutzler, C. Efficient photodissociation of anions from benzoyl-
functionalized ferrocene complexes. *Inorganic chemistry* **38**, 4861-4867 (1999).
- 21 Lovat, G. *et al.* Determination of the structure and geometry of N-heterocyclic carbenes
on Au (111) using high-resolution spectroscopy. *Chemical Science* **10**, 930-935 (2019).
- 22 Doud, E. A. *et al.* In situ formation of N-heterocyclic carbene-bound single-molecule
junctions. *Journal of the American Chemical Society* **140**, 8944-8949 (2018).
- 23 Ohnishi, H., Kondo, Y. & Takayanagi, K. Quantized conductance through individual rows
of suspended gold atoms. *Nature* **395**, 780-783 (1998).
- 24 Xu, B. & Tao, N. J. Measurement of single-molecule resistance by repeated formation of
molecular junctions. *science* **301**, 1221-1223 (2003).
- 25 Komoto, Y., Fujii, S., Iwane, M. & Kiguchi, M. Single-molecule junctions for molecular
electronics. *Journal of Materials Chemistry C* **4**, 8842-8858 (2016).
- 26 Aradhya, S. V. & Venkataraman, L. Single-molecule junctions beyond electronic
transport. *Nature nanotechnology* **8**, 399-410 (2013).
- 27 Capozzi, B. *et al.* Tunable charge transport in single-molecule junctions via electrolytic
gating. *Nano letters* **14**, 1400-1404 (2014).
- 28 Fujii, S. *et al.* Highly-conducting molecular circuits based on antiaromaticity. *Nature
Communications* **8**, 15984 (2017).
- 29 Vezzoli, A. *et al.* Gating of single molecule junction conductance by charge transfer
complex formation. *Nanoscale* **7**, 18949-18955 (2015).
- 30 Yin, X. *et al.* A reversible single-molecule switch based on activated antiaromaticity.
Science advances **3**, eaao2615 (2017).
- 31 Camarasa-Gómez, M. *et al.* Mechanically Tunable Quantum Interference in Ferrocene-
Based Single-Molecule Junctions. *Nano Letters* **20**, 6381-6386 (2020).
- 32 Li, S. *et al.* Using automated synthesis to understand the role of side chains on molecular
charge transport. *Nature communications* **13**, 2102 (2022).
- 33 Li, J. *et al.* Ladder-type conjugated molecules as robust multi-state single-molecule
switches. *Chem* (2023).
- 34 Fu, T. *et al.* Enhanced coupling through π -stacking in imidazole-based molecular
junctions. *Chemical Science* **10**, 9998-10002 (2019).
- 35 Adak, O. *et al.* Flicker noise as a probe of electronic interaction at metal–single molecule
interfaces. *Nano letters* **15**, 4143-4149 (2015).
- 36 Batra, A. *et al.* Tuning rectification in single-molecular diodes. *Nano letters* **13**, 6233-
6237 (2013).
- 37 Capozzi, B. *et al.* Single-molecule diodes with high rectification ratios through
environmental control. *Nature nanotechnology* **10**, 522-527 (2015).
- 38 Lin, J. *et al.* Cleavage of non-polar C (sp²)–C (sp²) bonds in cycloparaphenylenes via
electric field-catalyzed electrophilic aromatic substitution. *Nature Communications* **14**,
293 (2023).
- 39 Manrique, D. Z. *et al.* A quantum circuit rule for interference effects in single-molecule
electrical junctions. *Nature communications* **6**, 6389 (2015).
- 40 Harashima, T. *et al.* Single-molecule junction spontaneously restored by DNA zipper.
Nature Communications **12**, 5762 (2021).
- 41 Garra, P. *et al.* Ferrocene-based (photo) redox polymerization under long wavelengths.
Polymer Chemistry **10**, 1431-1441 (2019).

- 42 Shanmugam, S., Xu, J. & Boyer, C. Light-regulated polymerization under near-infrared/far-red irradiation catalyzed by bacteriochlorophyll A. *Angewandte Chemie* **128**, 1048-1052 (2016).
- 43 Kamenetska, M., Widawsky, J., Dell'Angela, M., Frei, M. & Venkataraman, L. Temperature dependent tunneling conductance of single molecule junctions. *The Journal of Chemical Physics* **146** (2017).

REVIEWER COMMENTS

Reviewer #1 (Remarks to the Author):

The authors provide additional data that, at least, convince me that their observations of the unusual conductance peak is reproducible. The formation of Au-Fe bond could be the most logical guess, although I still cannot figure out the chemical reaction here (maybe the reactive Au⁰ can be considered having a strong Lewis acidic property). Unfortunately, the extra chemistry proof data that I suggested is difficult for them to present and I understand the difficulty here. I suggest the author slightly weaken their conclusion saying that the Au-Fe bond is the most possible inference, and then I'm happy to accept the publication of this work.

Reviewer #2 (Remarks to the Author):

I appreciate the effort the authors have put in to respond to the reviewer comments and some improvement has clearly resulted from this. However, several comments have not been addressed in a satisfactory and therefore my overall assessment of the manuscript has not changed. Specifically, this refers to the following points (please refer to the response document for the authors' response):

1) "Given that that the authors already include solution-based voltammetry data, the latter should be relatively straightforward to obtain (as would XPS data, in fact)."

I don't understand the argument why adlayers of the molecule cannot be produced for this molecule, especially if the binding energy is about 0.8 eV. This is stronger than the pyridin-gold bond and (sub-)monolayers of this system have been characterised in great detail (surface electrochemistry, vibrational spectroscopy, surface spectroscopies such as XPS). Even if the lifetime of an individual bond relatively short, this does not mean that layers can be formed under appropriate conditions. Moreover, I was not suggesting that the authors obtain a crystal structure of a molecule in the junction ("However, it's important to note that obtaining crystal structure spectra of the molecules bound in the junction, linked by the Fe-Au bond, is currently unattainable either in situ or ex situ"). But if there is indeed such a relatively strong bond between the Fe centre and the Au surface and the authors already have solution-based CV data, then I don't see any reason why surface electrochemical measurements and also XPS data cannot be provided. Equally, the supposed absence of undercoordinated Au atoms is not a valid argument in my view, as the surface can also be roughened.

2) "Ultimately, electrochemical STM would be a desirable tool for the present study..."

Firstly, electrochemical STM involves a four-electrode setup (counter, reference and two working electrodes), not three. The main point about increased accuracy was not about increasing accuracy in the conductance measurements, but in electrochemical potential control. And while it is possible to run two-electrode experiments with good potential control (when the counter electrode is much larger, as then the potential drop at that interface is likely going to be small and constant), that configuration is clearly not the same. In fact, in a four-electrode configuration, gate and bias voltages can be controlled independently, allowing for conductance measurements at the same tip-substrate bias, but different electrochemical potentials. Also the comment that "We note that gating experiments cannot be performed on individual junctions ("hold" measurements) as it takes time for the electrolytes to reorganize around the junction" is simply incorrect and has been done by several groups over the last 10-15 years.

3) Hence, the manuscript requires substantial work...why did they only use the frequency range from 100-1000 Hz for their fit?).

The authors' response here, "We avoid the region above 1000 Hz to avoid any noise introduced by

our current amplifier at the 100 kHz acquisition rate", is unsatisfactory. Perhaps they do struggle with line frequency noise, even though I would note that impedance measurements are routinely done from 1 Hz to 100 KHz, but why would a 100 kHz acquisition rate limit the measurement to 1 kHz? It would seem perfectly feasible to go up to 10 or 20 kHz, taking into account filtering as well.

4) "Finally, there is at least one statement, which may not be correct and I would like the authors to explain further, namely that in a unpolar solvent the redox state of a molecule can not be changed due to changes in bias (p. 5). In the total absence of ions at the interface (which may be experimentally unrealistic, even in unpolar solvents), one would expect the potential drop between substrate and tip to be linear..."

Firstly, the authors' make the point that the lack of voltammetric response supposedly shows that the molecule cannot be switched. I don't think one can draw this conclusion, because it could simply be that the solution concentration and the solubility of the molecule is too low to produce a measurable voltammetric response. However, this does not mean that the molecule cannot be switched in a junction, where the potential drop is linear (in the absence of a double layer) or of a more complex nature (in the presence of ions). Notably, a surface electrochemical measurement would potentially address this, as the surface concentration is determined by the coverage (see above). Indeed, as the authors point out above, the potential drop at the redox site is likely asymmetric, due to the asymmetry in the electroactive areas of the electrodes. So I don't see any reason why in principle the local potential at the redox site cannot be changed by changing the applied bias, even in the absence of ions. The lack of electrochemical (bipotentiostatic) control of course means that offset between the Fermi levels of the electrodes and the molecular energy levels is unknown in their experimental configuration and cannot be controlled directly.

So overall, I am not satisfied with the response to all of the points I had raised in relation to the previous version of the manuscript, even though I do appreciate that some improvements have been made.

Reviewer #3 (Remarks to the Author):

The authors have very carefully revised their manuscript. As a whole, it is now much better organized and results in clear conclusions that are based on their experimental and theoretical findings.

The results are interesting and novel. Based on their set of investigations, I see sufficient indications for the correctness of the conclusions drawn by the authors so that I recommend publication of this paper in its present form.

It is obvious that there still many open questions, in part raised by the referees, but most of them clearly are far beyond the scope of this paper.

REVIEWER COMMENTS

Reviewer #1 (Remarks to the Author):

The authors provide additional data that, at least, convince me that their observations of the unusual conductance peak is reproducible. The formation of Au-Fe bond could be the most logical guess, although I still cannot figure out the chemical reaction here (maybe the reactive Au⁰ can be considered having a strong Lewis acidic property). Unfortunately, the extra chemistry proof data that I suggested is difficult for them to present and I understand the difficulty here. I suggest the author slightly weaken their conclusion saying that the Au-Fe bond is the most possible inference, and then I'm happy to accept the publication of this work.

Response: Thank you for your comprehensive review, valuable suggestions and recommendation that our work is accepted for publication.

We first want to state that we are open to weaken our conclusion as this reviewer suggests but before doing this, we would like the reviewer to consider some additional data that we have gathered.

In order to corroborate our conclusions, we conducted the additional STM-BJ measurements of group 8 metallocenes—ferrocene, ruthenocene, and osmocene. We have measured conductances of these three metallocenes at biases sufficiently high to generate the oxidized species. **Figure R9** shows that only osmocene exhibits a conductance peak, involving the formation of a junction between two gold electrodes.

Figure R9. Overlaid one-dimensional (1D) conductance histograms of group 8 metallocenes measured in propylene carbonate (PC) at bias voltages of 700 mV for ferrocene and ruthenocene, and 500 mV for osmocene to ensure that they are oxidized.

We can conclude from these results the following:

1. All three molecules have very similar sizes, with the distance between two Cp rings varying by less than 0.30 Å.¹⁻³ This difference is negligible when forming a molecular junction in the STM-BJ setup. Therefore, the fact that we don't see junctions with ferrocene cannot be attributed to its small size.

2. If junctions that involve just Cp-Au bonds were more likely to form than the metal center (Fe, Ru, and Os)-Au bond, both ferrocene and ruthenocene, possessing Cp rings on both sides, should exhibit similar peaks. However, only osmocene forms such a junction.
3. The metal center-Au bond must be forming in our measurements.

To explain the formation of the osmocene junction between two gold electrodes, we propose that one bond is initially formed through Os-Au, followed by the second bond formation through osmocene moiety. Notably, owing to the slightly larger size of the osmium atom, osmocene is susceptible to ring slippage during interactions with ligands at the metal center (η^5 to η^3 or η^1 Cp ligand coordination)—a phenomenon not typically observed in other metallocenes.⁴

Consequently, we attribute the mechanism for the second bond formation to the ring slippage phenomenon. Since such a ring slippage is not expected in ruthenocene or ferrocene, even if the Au-binds to the metal center we will not form a junction as a second bond to the other electrode cannot be formed. This is work that is still in progress. We do not have a comprehensive explanation of how ring slippage occurs and how osmocene binds to gold but this is in any case beyond the scope of the present manuscript.

In summary, our findings strongly indicate that the formation of the Fe-Au bond is much more likely than a junction formed through a Cp-Au bond. We believe this manuscript will serve as the foundation for future studies on metallocene-based molecular junctions.

Reviewer #2 (Remarks to the Author):

I appreciate the effort the authors have put into to respond to the reviewer comments and some improvement has clearly resulted from this. However, several comments have not been addressed in a satisfactory and therefore my overall assessment of the manuscript has not changed. Specifically, this refers to the following points (please refer to the response document for the authors' response):

1) "Given that that the authors already include solution-based voltammetry data, the latter should be relatively straightforward to obtain (as would XPS data, in fact)."

I don't understand the argument why adlayers of the molecule cannot be produced for this molecule, especially if the binding energy is about 0.8 eV. This is stronger than the pyridine-gold bond and (sub-)monolayers of this system have been characterised in great detail (surface electrochemistry, vibrational spectroscopy, surface spectroscopies such as XPS). Even if the lifetime of an individual bond relatively short, this does not mean that layers can be formed under appropriate conditions. Moreover, I was not suggesting that the authors obtain a crystal structure of a molecule in the junction ("However, it's important to note that obtaining crystal structure spectra of the molecules bound in the junction, linked by the Fe-Au bond, is currently unattainable either in situ or ex situ"). But if there is indeed such a relatively strong bond between the Fe centre and the Au surface and the authors already have solution-based CV data, then I don't see any reason why surface electrochemical measurements and also XPS data cannot be provided. Equally, the supposed absence of undercoordinated Au atoms is not a valid argument in my view, as the surface can also be roughened.

Response: We beg to differ. Many studies have reported the binding energy of pyridine-gold bonding which range from 0.8 to 1.4 eV without considering van der Waals (vdW) interaction.⁵⁻⁸ We also calculated the binding energy between pyridine and gold electrode using the same procedure and software outlined in our manuscript. The resulting energy is 0.80 eV, excluding vdW, which is significantly higher than that of Fe-Au bonding, 0.42 eV (without vdW). Moreover, in our previous study, we experimentally determined the binding energy of pyridine-gold bonding to be ~0.9-1.0 eV using near edge X-ray absorption fine structure (NEXAFS) spectroscopy as most pyridine desorbed from the gold surface around 273 K.⁹ This implies that the Fe-Au bond is too weak to create monolayers on Au at room temperature for x-ray spectroscopy studies. Utilizing the Redhead formula (Eq. R1), we can compute the desorption temperature of ferrocene on Au when bound through an Fe-Au bond.

$$E_d = k_B T_p \ln\left(\frac{\nu T_p}{\beta}\right) \quad \text{Eq. R1}$$

Here, E_d is the binding energy, k_B is the Boltzmann constant, T_p is the desorption temperature, ν is a pre-factor of 10^{13} s^{-1} , β and is a heating rate.¹⁰ If the desorption temperature is 300 K at a typical heating rate β of 1 K/s, the binding energy needs to be 0.9 eV. Conversely, for a binding energy of 0.8 eV, the Redhead formula gives a desorption temperature of 260 K. Consequently, it is impractical to prepare the Fe-Au bonding samples by surface electrochemistry at room temperatures. While it is conceivable to maintain the temperature below 260 K to preserve the Fe-Au bond, such low temperatures are not suitable for conducting electrochemical studies in solution. This is due to potential issues such as changes in diffusion constants, analyte/electrolyte precipitation, and ice formation unless the experiments are conducted in absence of water. Therefore, characterizing the Fe-Au bond using XPS is not currently feasible.

Finally, given our results with osmocene, presented in response to Reviewer 1's comments, we attempted to carry out XPS measurements of osmocene deposited on Au since we experimentally found that it forms junctions both in the neutral state and the oxidized state in the STM-BJ setup. We are unfortunately not able to detect the XPS peaks from osmocene in our room temperature Phi 5500 XPS instrument. We find that the osmocene desorbs from the Au substrate once inserted into the vacuum chamber.

[Redacted]

2) *"Ultimately, electrochemical STM would be a desirable tool for the present study..."*

Firstly, electrochemical STM involves a four-electrode setup (counter, reference and two working electrodes), not three. The main point about increased accuracy was not about increasing accuracy in the conductance measurements, but in electrochemical potential control. And while it is possible to run two-electrode experiments with good potential control (when the counter electrode is much larger, as then the potential drop at that interface is likely going to be small and constant), that configuration is clearly not the same. In fact, in a four-electrode configuration, gate and bias voltages can be controlled independently, allowing for conductance measurements at the same tip-substrate bias, but different electrochemical potentials. Also the comment that "We note that gating experiments cannot be performed on individual junctions ("hold" measurements) as it takes time for the electrolytes to reorganize around the junction" is simply incorrect and has been done by several groups over the last 10-15 years.

Response: We agree that a conventional electrochemical STM has four electrodes. However, in STM-BJ setups, we and others have demonstrated that a three-electrode system operates equivalently to the conventional four-electrode system over the last decade.¹¹⁻¹³ To elaborate, we can employ a gate electrode that functions as both the counter and reference electrodes, while the gold tip and substrate serve as the working electrodes. Hence, an electrochemical STM setup does not inherently require four electrodes as long as we calibrate the potential using the redox events of ferrocene, as shown in **Fig. R11**. Furthermore, the two-electrode system in the STM-BJ setup has been thoroughly verified for its precise control of electrochemical potential.¹⁴⁻¹⁷ In the previous rebuttal, **Fig. R5** clearly demonstrated the accuracy in electrochemical potential control of a two-electrode system. The molecule **1** undergoes a change in its oxidation state at ~300 mV, resulting in distinct junction geometries—either **1H** or **1L**—with a corresponding shift in the conductance

peak. Moreover, the broaden peak (or two-peak) at around the redox potential indicates the Nernstian-type behavior around the redox potential as the review suggested. These are consistent with our observations throughout the manuscript including in-situ CV measurements (**Fig. S1**).

[Redacted]

Nonetheless, we conducted three-electrode STM-BJ measurements, employing a Pt wire gate electrode that is in the solution of **1** with TBAPF₆ as electrolyte. To ensure that the applied gate potential drops across the working electrodes, we use a small tip/substrate bias of 50 mV. We varied the gate potential voltage and measured conductance traces. As the gating voltage becomes more negative, more molecules in the solution are oxidized at the tip and substrate electrodes. We observe in the higher-conducting and shorter junction (**1H**), as depicted in **Fig. R12**. Note that the potential applied to the gate necessary to oxidize the molecule is higher than the potential applied to the tip in the 2-electrode system as the gating efficiency is different in these two setups as discussed previously.¹¹ It is also worth noting the presence of two peaks around the gate potential of 1 V, indicating Nernstian-type behavior. These results are essentially the same as findings from the two-electrode STM-BJ measurements.

As you can see, these gating measurements basically replicate our light-induced oxidation experiments (**Fig. 1**), chemical oxidant measurements (**Fig. S5**) and the two-electrode measurements. We do not believe that adding these electrochemical gating data to the manuscript will not provide any additional information however, if this reviewer insists on adding these new data, we can provide them in the SI.

3) Hence, the manuscript requires substantial work...why did they only use the frequency range from 100-1000 Hz for their fit?).

The authors' response here, "We avoid the region above 1000 Hz to avoid any noise introduced by our current amplifier at the 100 kHz acquisition rate", is unsatisfactory. Perhaps they do struggle with line frequency noise, even though I would note that impedance measurements are routinely done from 1 Hz to 100 KHz, but why would a 100 kHz acquisition rate limit the measurement to 1 kHz? It would seem perfectly feasible to go up to 10 or 20 kHz, taking into account filtering as well.

Response: We acknowledge the feasibility of extending the frequency range for analysis, given that we measure flicker noise at 100 kHz. Typically, our research group and others analyze within the range of 0.1–1 kHz, which has proven sufficient to yield meaningful results for flicker noise analysis.¹⁸⁻²¹ However, to address the reviewer comment, we provide supplementary analysis results using the range of 1–10 kHz (**Fig. R13**). The resulting 2D histogram remains almost the same as that in **Fig. 4** of the main manuscript. The exponent values of n exhibit slight changes from 1.28 to 1.20 (**1L**) and from 1.30 to 1.27 (**1H**) and the conclusion remains the same, i.e. that both junctions are through-bond coupled.

[Redacted]

4) "Finally, there is at least one statement, which may not be correct and I would like the authors to explain further, namely that in a unpolar solvent the redox state of a molecule can not be changed due to changes in bias (p. 5). In the total absence of ions at the interface (which may be experimentally unrealistic, even in unpolar solvents), one would expect the potential drop between substrate and tip to be linear..."

Firstly, the authors' make the point that the lack of voltammetric response supposedly shows that the molecule cannot be switched. I don't think one can draw this conclusion, because it could simply be that the solution concentration and the solubility of the molecule is too low to produce a measurable voltammetric response. However, this does not mean that the molecule cannot be switched in a junction, where the potential drop is linear (in the absence of a double layer) or of a more complex nature (in the presence of ions). Notably, a surface electrochemical measurement would potentially address this, as the surface concentration is determined by the coverage (see above). Indeed, as the authors point out above, the potential drop at the redox site is likely asymmetric, due to the asymmetry in the electroactive areas of the electrodes. So I don't see any reason why in principle the local potential at the redox site cannot be changed by changing the applied bias, even in the absence of ions. The lack of electrochemical (bipotentiostatic) control of course means that offset between the Fermi levels of the electrodes and the molecular energy levels is unknown in their experimental configuration and cannot be controlled directly.

Response: To oxidize a molecule in a solution, we would need to increase the applied bias voltage beyond the redox potential of the molecule. In the STM-BJ setup using a non-polar solvent, the potential drop is symmetric between the tip and substrate.²² In contrast, in polar systems, we asymmetrically open the bias window by coating the tip surface, facilitating oxidation of the molecule.²³ Theoretically, applying a bias voltage two times higher than the redox potential in a non-polar solvent could oxidize molecules. However, as the reviewer noted, the concentration of stabilized oxidized species in the system would be very low due to the absence of surrounding ions.

In addition, it is energetically unfavorable to form charged species in non-polar solvents and this will likely result in an increase in the redox potential of the molecule in non-polar solvents. Consequently, the probability of oxidized molecule junction formation would be very small. Pursuing this approach seems to offer no clear benefits.

Additionally, we acknowledge that our statement was not correct and thank the reviewer for pointing this out. What we meant to add was that in non-polar solvents without supporting electrolytes, the redox state of a molecule cannot be changed by changes in bias voltage within a reasonable bias window. We have clarified this point in the main manuscript. (Line 102) It is worth noting that there is generally no expectation for researchers to conduct CV measurements in non-polar solvents without electrolytes, reflecting common practices in the field. While it might be possible to observe the oxidized molecular junction by measuring **1** in a non-polar solvent and applying large tip bias voltages, this junction will be bound through two Au-SMe links and we are unlikely to convert this junction to one that binds through one Au-Fe bond and one Au-SMe link. Since the concentration of oxidized molecules in the solution at high bias is likely to be very small (with a majority of the molecules being in a neutral state), we will not observe the high conductance peak.

So overall, I am not satisfied with the response to all of the points I had raised in relation to the previous version of the manuscript, even though I do appreciate that some improvements have been made.

Response: We hope that the explanations and evidence presented above have addressed the reviewer's concerns to their satisfaction.

Reviewer #3 (Remarks to the Author):

The authors have very carefully revised their manuscript. As a whole, it is now much better organized and results in clear conclusions that are based on their experimental and theoretical findings.

The results are interesting and novel. Based on their set of investigations, I see sufficient indications for the correctness of the conclusions drawn by the authors so that I recommend publication of this paper in its present form.

It is obvious that there still many open questions, in part raised by the referees, but most of them clearly are far beyond the scope of this paper.

Response: We appreciate your thorough review, valuable suggestions and recommending publication. To address the open questions, we have conducted a study involving a series of group 8 metallocenes, as mentioned above in response to Reviewer 1's comments.

References

- 1 Hardgrove, G. L. & Templeton, D. H. The crystal structure of ruthenocene. *Acta Crystallographica* **12**, 28-32 (1959).
- 2 Bohn, R. K. & Haaland, A. On the molecular structure of ferrocene, Fe (C₅H₅)₂. *Journal of Organometallic Chemistry* **5**, 470-476 (1966).
- 3 Bobyens, J., Levendis, D., Bruce, M. I. & Williams, M. L. Crystal structure of osmocene, Os (η-C₅H₅)₂. *Journal of Crystallographic and Spectroscopic Research* **16**, 519-524 (1986).
- 4 Cheung, W.-M. *et al.* Facile η⁵–η¹ Ring Slippage of the Cycloolefin Ligands in Osmocene and Bis (η⁵-indenyl) ruthenium (II). *Inorganic Chemistry* **52**, 10449-10455 (2013).
- 5 Bilić, A., Reimers, J. R. & Hush, N. S. Adsorption of pyridine on the gold (111) surface: implications for “alligator clips” for molecular wires. *The Journal of Physical Chemistry B* **106**, 6740-6747 (2002).
- 6 Mollenhauer, D., Flob, J., Reissig, H. U., Voloshina, E. & Paulus, B. Accurate quantum-chemical description of gold complexes with pyridine and its derivatives. *Journal of Computational Chemistry* **32**, 1839-1845 (2011).
- 7 Mollenhauer, D., Gaston, N., Voloshina, E. & Paulus, B. Interaction of pyridine derivatives with a gold (111) surface as a model for adsorption to large nanoparticles. *The Journal of Physical Chemistry C* **117**, 4470-4479 (2013).
- 8 Wu, D., Hayashi, M., Chang, C., Liang, K. & Lin, S. Bonding interaction, low-lying states and excited charge-transfer states of pyridine–metal clusters: Pyridine–M_n (M= Cu, Ag, Au; n= 2–4). *The Journal of chemical physics* **118**, 4073-4085 (2003).
- 9 Cvetko, D. *et al.* Ultrafast electron injection into photo-excited organic molecules. *Physical Chemistry Chemical Physics* **18**, 22140-22145 (2016).
- 10 Redhead, P. A. Thermal desorption of gases. *vacuum* **12**, 203-211 (1962).
- 11 Capozzi, B. *et al.* Tunable charge transport in single-molecule junctions via electrolytic gating. *Nano letters* **14**, 1400-1404 (2014).
- 12 Tian, J. H. *et al.* Electrochemically assisted fabrication of metal atomic wires and molecular junctions by MCBJ and STM-BJ methods. *ChemPhysChem* **11**, 2745-2755 (2010).
- 13 Zhou, P. *et al.* Electrostatic gating of single-molecule junctions based on the STM-BJ technique. *Nanoscale* **13**, 7600-7605 (2021).
- 14 Dell, E. J., Capozzi, B., Xia, J., Venkataraman, L. & Campos, L. M. Molecular length dictates the nature of charge carriers in single-molecule junctions of oxidized oligothiophenes. *Nature chemistry* **7**, 209-214 (2015).
- 15 Greenwald, J. E. *et al.* Highly nonlinear transport across single-molecule junctions via destructive quantum interference. *Nature Nanotechnology* **16**, 313-317 (2021).
- 16 Li, S. *et al.* Characterizing intermolecular interactions in redox-active pyridinium-based molecular junctions. *Journal of Electroanalytical Chemistry* **875**, 114070 (2020).
- 17 Wang, Z. *et al.* Electrochemically controlled rectification in symmetric single-molecule junctions. *Proceedings of the National Academy of Sciences* **119**, e2122183119 (2022).
- 18 Adak, O. *et al.* Flicker noise as a probe of electronic interaction at metal–single molecule interfaces. *Nano letters* **15**, 4143-4149 (2015).

- 19 Fu, T. *et al.* Enhanced coupling through π -stacking in imidazole-based molecular junctions. *Chemical Science* **10**, 9998-10002 (2019).
- 20 Li, S. *et al.* Using automated synthesis to understand the role of side chains on molecular charge transport. *Nature communications* **13**, 2102 (2022).
- 21 Pan, Z. *et al.* The Evolution of the Charge Transport Mechanism in Single-Molecule Break Junctions Revealed by Flicker Noise Analysis. *Small* **18**, 2107220 (2022).
- 22 Darancet, P., Widawsky, J. R., Choi, H. J., Venkataraman, L. & Neaton, J. B. Quantitative current–voltage characteristics in molecular junctions from first principles. *Nano letters* **12**, 6250-6254 (2012).
- 23 Fung, E.-D. *et al.* Breaking down resonance: Nonlinear transport and the breakdown of coherent tunneling models in single molecule junctions. *Nano letters* **19**, 2555-2561 (2019).

REVIEWERS' COMMENTS

Reviewer #2 (Remarks to the Author):

I would like to thank the authors for their response, to which I have added the following comments:

A) "Response: We beg to differ. Many studies have reported the binding energy of pyridine-gold bonding..."

I appreciate the discussion around the Redhead formula, which is however about thermal desorption into vacuum. Arguably, this applies to considerations around the proposed XPS study, but for the electrochemical experiment, the physical picture is more accurately described by an adsorption equilibrium from solution, which under the right experimental conditions could make the system amenable to surface-electrochemical or surface-sensitive spectroscopic investigation. Nevertheless, I thank the authors for addressing this point in more detail now.

B) "...we and others have demonstrated that a three-electrode system operates equivalently to the conventional four-electrode system over the last decade.¹¹⁻¹³ To elaborate, we can employ a gate electrode that functions as both the counter and reference electrodes, while the gold tip and substrate serve as the working electrodes. Hence, an electrochemical STM setup does not inherently require four electrodes as long as we calibrate the potential using the redox events of ferrocene, as shown in Fig. R11."

The reason for separating counter and reference electrodes in an electrochemical environment is that the counter electrode can carry sufficient current to establish the potential drop between the reference (gate) electrode and the working electrode(s). At the same time, the reference electrode should not carry significant currents, so that the potential drop at its interface with the solution is unaffected and as constant as possible. This is the principal motivation behind using potentiostatic potential control and combining the two will only work well, if the current drop the electrochemical cell is small. Arguably, this will be the case when there are no redox-active species in solution and no (or little) Faradaic processes taking place at the electrodes. In the present case, there does seem to be electroactive species in solution. However, more generally a configuration where counter and reference electrodes are separated will provide more accurate potential control, compared when those two electrodes are combined. I didn't think that this necessarily invalidated the conclusions by the authors, but I felt the point needed to be made.

C) "Moreover, the broaden peak (or two-peak) at around the redox potential indicates the Nernstian-type behavior around the redox potential as the review suggested."

Just because there are two peaks does not mean that the behavior is Nernstian. One would have to show that the ratio of oxidized vs. reduced species follows the Nernst equation, which I don't think has been done yet.

D) "To ensure that the applied gate potential drops across the working electrodes, we use a small tip/substrate bias of 50 mV."

I don't understand the argument here. There is a potential drop at the solution/gate electrode interface and there are potential drops at the interfaces between the solution and the two working electrodes, noting that the latter two likely overlap in the junction while most of the electrode surface is outside of this region. The characteristic decay length is typically very small for moderate electrolyte concentrations and far shorter than the typically macroscopic distance between the RE/CE and working electrodes. Could the authors please clarify what they mean here?

E) "Figure R11. (a) In situ cyclic voltammetry (CV) of 50 μM ferrocene in propylene carbonate (PC) using two-electrode STM-BJ setup. Note that since we are using a microelectrode without electrolyte, we do not observe the standard "duck" shaped peaks as the oxidation process is not diffusion limited."

Figure R11 clearly shows diffusion-limited behaviour, except that it is not linear diffusion (as for a "duck shaped" CV, for want of a better term), but hemispherical diffusion (as one would indeed expect for a microelectrode). What is notable here is that the current does not plateau at high potentials, which is expected for CV at a micro electrode in the mass transport-controlled limit. I wonder whether the authors have an explanation for this behavior.

F) "The authors' response here, "We avoid the region above 1000 Hz to avoid any noise introduced by our current amplifier at the 100 kHz acquisition rate", is unsatisfactory. Perhaps they do struggle with line frequency noise, even though I would note that impedance measurements are routinely done from 1 Hz to 100 KHz, but why would a 100 kHz acquisition rate limit the measurement to 1 kHz? It would seem perfectly feasible to go up to 10 or 20 kHz, taking into account filtering as well."

Thanks for addressing this, looks fine now.

G) In relation to point 4:

"In contrast, in polar systems, we asymmetrically open the bias window by coating the tip surface, facilitating oxidation of the molecule.²³ Theoretically, applying a bias voltage two times higher than the redox potential in a non-polar solvent could oxidize molecules."

This does not make any sense to me. Why would coating the electrode facilitate the oxidation of a molecule? In the current-free case, the potential drop at the electrode solution interface is defined by the ion distribution at the interface, which will not change by changing the electroactive area of the electrode. If there is current flow, then a reduction of the electroactive area (at a well-controlled electrochemical potential) will result in a reduction of the current, while the current density and hence interfacial potential drop remain the same. The second statement above is currently unsupported and rather speculative, from my point of view.

"It is worth noting that there is generally no expectation for researchers to conduct CV measurements in nonpolar solvents without electrolytes, reflecting common practices in the field."

The proposed experiment does not strike me as a prohibitively challenging one and I would argue that while there is no "general expectation" to do such measurements, I have argued that they are needed to support some of the claims made in the paper. If this support can be provided by other means, that is acceptable - otherwise the claims would have to be modified.

Reviewer #3 (Remarks to the Author):

In my previous report I already recommended publication of this paper essentially in its present form. Now going through the comments and partly further requests by the other referees my opinion did not change for the following reasons:

1) Reviewer #1 basically accepted publication and only recommends softening of the conclusions since the chemical process leading to formation of the suggested chemical bond between Fe and Au could not yet been clarified. The authors respond by showing a new example of osmocene that exhibits similar behavior in the oxidized state as the ferrocene derivative under investigation here, in contrast to pure ferrocene and ruthenocene. Indeed, this observation is very interesting, although the underlying mechanism still remains to be clarified. It will clearly trigger further investigations of bond and contact formation in this class of molecules, and may be taken as an indication that the behavior reported here does not represent a unique case.

2) Reviewer #2 raised several points of criticism:

a) He/she insists on the feasibility of XPS spectra of the molecular layer on the Au substrate that the authors could not provide. He/she argues that XPS should be able to prove the existence of

chemical bond formation between Au and Fe in the oxidized state of the ferrocene unit, under which the high conductance can only be observed. This argument, however, is highly questionable, since the XPS spectra have to be taken ex-situ in a dry environment. The oxidized state obviously requires adjustment of the local chemical potential to a value that differs significantly from zero. Therefore, under zero bias only a small minority of the desired species can be expected. For non-zero bias I see no straightforward way how this condition could be reliably realized in an XPS experiment. Even if one neglects the experimental difficulties mentioned by the authors, this means that the XPS experiment will mainly detect the neutral species on flat Au surfaces that is condensed by van-der-Waals forces. It is true that bond formation, particularly on gold, depends strongly on local coordination and on particle size, but the suggestion by the referee of just making a „rough“ surface (on what length scales?) lacks the necessary precision required for yielding a clear-cut answer to the problem. Even if successful, it would again, for the reason just mentioned, generate (several) minority species that are hard to detect. Therefore, I don't think that this suggestion will be helpful in solving the problem.

b) Lack of precision of measurements: The authors convincingly show that a three-electrode experiment reproduces the transition from neutral to the oxidized state and the associated change of displacement at which the high conductance can be observed. I agree with the referee that the two-electrode measurements, even after calibration, are not as precise as those with three or four electrodes, but the authors demonstrate that the basic qualitative behavior is the same, independent of the setup used. Since this new result adds to the reliability of the data analysis, I suggest that these data will be added to the supplement.

c) Flicker noise data were extended to the 1 to 10 kHz range and shown to be, within error bars, independent of the range of frequencies used. Therefore, I consider this analysis to be valid.

d) Influence of polarizability of the surrounding electrolyte molecules. This discussion resulted in the correction of an error in the manuscript, but otherwise no new aspects were raised.

Therefore, with the small addition to the supplement, I recommend publication of this paper as is.

Reviewer #2 (Remarks to the Author):

I would like to thank the authors for their response, to which I have added the following comments:

A) "Response: We beg to differ. Many studies have reported the binding energy of pyridine-gold bonding..."

I appreciate the discussion around the Redhead formula, which is however about thermal desorption into vacuum. Arguably, this applies to considerations around the proposed XPS study, but for the electrochemical experiment, the physical picture is more accurately described by an adsorption equilibrium from solution, which under the right experimental conditions could make the system amenable to surface-electrochemical or surface-sensitive spectroscopic investigation. Nevertheless, I thank the authors for addressing this point in more detail now.

Response: We presented the Redhead formula exclusively to address the reviewer's comment. The Redhead formula substantiates that characterizing the Fe-Au bond using XPS is not feasible at room temperature in vacuum.

B) "...we and others have demonstrated that a three-electrode system operates equivalently to the conventional four-electrode system over the last decade.11-13 To elaborate, we can employ a gate electrode that functions as both the counter and reference electrodes, while the gold tip and substrate serve as the working electrodes. Hence, an electrochemical STM setup does not inherently require four electrodes as long as we calibrate the potential using the redox events of ferrocene, as shown in Fig. R11."

The reason for separating counter and reference electrodes in an electrochemical environment is that the counter electrode can carry sufficient current to establish the potential drop between the reference (gate) electrode and the working electrode(s). At the same time, the reference electrode should not carry significant currents, so that the potential drop at its interface with the solution is unaffected and as constant as possible. This is the principal motivation behind using potentiostatic potential control and combining the two will only work well, if the current drop the electrochemical cell is small. Arguably, this will be the case when there are no redox-active species in solution and no (or little) Faradaic processes taking place at the electrodes. In the present case, there does seem to be electroactive species in solution. However, more generally a configuration where counter and reference electrodes are separated will provide more accurate potential control, compared when those two electrodes are combined. I didn't think that this necessarily invalidated the conclusions by the authors, but I felt the point needed to be made.

Response: Thank you for addressing this point and clarifying your comment.

C) "Moreover, the broaden peak (or two-peak) at around the redox potential indicates the Nernstian-type behavior around the redox potential as the review suggested."

Just because there are two peaks does not mean that the behavior is Nernstian. One would have to show that the ratio of oxidized vs. reduced species follows the Nernst equation, which I don't think has been done yet.

Response: We agree. Having two peaks is suggestive of Nernstian behavior. In effect, what we find is that both species can form junctions and be measured when the bias is around 0.5-1 V. However, we are not quantitatively verifying an exact adherence to the Nernstian equation.

D) "To ensure that the applied gate potential drops across the working electrodes, we use a small tip/substrate bias of 50 mV."

I don't understand the argument here. There is a potential drop at the solution/gate electrode interface and there are potential drops at the interfaces between the solution and the two working electrodes, noting that the latter two likely overlap in the junction while most of the electrode surface is outside of this region. The characteristic decay length is typically very small for moderate electrolyte concentrations and far shorter than the typically macroscopic distance between the RE/CE and working electrodes. Could the authors please clarify what they mean here?

Response: Ideally, the potential bias between the tip and substrate should be zero since they are both working electrodes and must thus be at the same potential. However, in our case, a bias voltage between the tip and substrate is necessary to measure the conductance of a molecular junction. By minimizing the tip/substrate bias, especially when compared to the bias between the gate electrode and tip/substrate electrode, the impact of the tip/substrate bias can be neglected. [Capozzi et al, *Nano Letters*, 2014]

E) "Figure R11. (a) In situ cyclic voltammetry (CV) of 50 μ M ferrocene in propylene carbonate (PC) using two-electrode STM-BJ setup. Note that since we are using a microelectrode without electrolyte, we do not observe the standard "duck" shaped peaks as the oxidation process is not diffusion limited."

Figure R11 clearly shows diffusion-limited behaviour, except that it is not linear diffusion (as for a "duck shaped" CV, for want of a better term), but hemispherical diffusion (as one would indeed expect for a microelectrode). What is notable here is that the current does not plateau at high potentials, which is expected for CV at a micro electrode in the mass transport-controlled limit. I wonder whether the authors have an explanation for this behavior.

Response: The plateau as noted by the reviewer for microelectrodes results from a limiting case only observed at very small electrodes and low scan rates, far from potentials where other redox processes may occur. Certainly, the exact form of the current-potential response may vary, leading us to expect a continuum of CV shapes between the "duck" and plateau curves, depending on both electrode size and scan rate, as noted in [Forster et al, *Handbook of electrochemistry* 2007]. Furthermore, these voltammograms are obtained in air, and beyond the oxidation of ferrocene we approach the limits of our potential window in oxygen-saturated, non-anhydrous, propylene carbonate. *In situ* cyclic voltammetry (CV) measurement of propylene carbonate demonstrates oxidation processes of the medium, as depicted in **Figure R13**. The increasing current we observe after ferrocene oxidation could therefore also be tentatively attributed to background oxidation processes that take place in this medium under our experimental conditions.

Figure R13. *In situ* cyclic voltammetry (CV) of propylene carbonate (PC) using two-electrode STM-BJ setup.

F) "The authors' response here, "We avoid the region above 1000 Hz to avoid any noise introduced by our current amplifier at the 100 kHz acquisition rate", is unsatisfactory. Perhaps they do struggle with line frequency noise, even though I would note that impedance measurements are routinely done from 1 Hz to 100 KHz, but why would a 100 kHz acquisition rate limit the measurement to 1 kHz? It would seem perfectly feasible to go up to 10 or 20 kHz, taking into account filtering as well."

Thanks for addressing this, looks fine now.

Response: Thank you for your acknowledgement.

G) In relation to point 4:

"In contrast, in polar systems, we asymmetrically open the bias window by coating the tip surface, facilitating oxidation of the molecule.²³ Theoretically, applying a bias voltage two times higher than the redox potential in a non-polar solvent could oxidize molecules."

This does not make any sense to me. Why would coating the electrode facilitate the oxidation of a molecule? In the current-free case, the potential drop at the electrode solution interface is defined by the ion distribution at the interface, which will not change by changing the electroactive area of the electrode. If there is current flow, then a reduction of the electroactive area (at a well-controlled electrochemical potential) will result in a reduction of the current, while the current density and hence interfacial potential drop remain the same. The second statement above is currently unsupported and rather speculative, from my point of view.

"It is worth noting that there is generally no expectation for researchers to conduct CV measurements in nonpolar solvents without electrolytes, reflecting common practices in the field."

The proposed experiment does not strike me as a prohibitively challenging one and I would argue that while there is no "general expectation" to do such measurements, I have argued that they are

needed to support some of the claims made in the paper. If this support can be provided by other means, that is acceptable - otherwise the claims would have to be modified.

Response: The reviewer's comment might hold true if the two electrodes were of the same shape and size. However, our tip and substrate have distinctly different shapes and surface areas. The tip is a nanoscale tip, whereas the substrate is a planar disc with a surface area of 0.8 cm². Furthermore, after coating only the tip, their surface area difference increases significantly. In our setup with a flowing current, the electric double layer at the exposed surface is much denser at the tip compared to the large planar gold substrate. We have discussed this in detail in an earlier work by Capozzi et al, *Nature Nanotechnology* 2015. Additionally, we have already presented STM-BJ measurements of **1** in a non-polar solvent in the Supplementary Information, and CV measurements of **1** in a non-polar solvent were provided in the first rebuttal. In both experiments, no oxidized state was detected.

Reviewer #3 (Remarks to the Author):

In my previous report I already recommended publication of this paper essentially in its present form. Now going through the comments and partly further requests by the other referees my opinion did not change for the following reasons:

1) Reviewer #1 basically accepted publication and only recommends softening of the conclusions since the chemical process leading to formation of the suggested chemical bond between Fe and Au could not yet been clarified. The authors respond by showing a new example of osmocene that exhibits similar behavior in the oxidized state as the ferrocene derivative under investigation here, in contrast to pure ferrocene and ruthenocene. Indeed, this observation is very interesting, although the underlying mechanism still remains to be clarified. It will clearly trigger further investigations of bond and contact formation in this class of molecules, and may be taken as an indication that the behavior reported here does not represent a unique case.

Response: We appreciate your acknowledgement of our study.

2) Reviewer #2 raised several points of criticism:

a) He/she insists on the feasibility of XPS spectra of the molecular layer on the Au substrate that the authors could not provide. He/she argues that XPS should be able to prove the existence of chemical bond formation between Au and Fe in the oxidized state of the ferrocene unit, under which the high conductance can only be observed. This argument, however, is highly questionable, since the XPS spectra have to be taken ex-situ in a dry environment. The oxidized state obviously requires adjustment of the local chemical potential to a value that differs significantly from zero. Therefore, under zero bias only a small minority of the desired species can be expected. For non-zero bias I see no straightforward way how this condition could be reliably realized in an XPS experiment. Even if one neglects the experimental difficulties mentioned by the authors, this means that the XPS experiment will mainly detect the neutral species on flat Au surfaces that is condensed by van-der-Waals forces. It is true that bond formation, particularly on gold, depends strongly on local coordination and on particle size, but the suggestion by the referee of just making a „rough“ surface (on what length scales?) lacks the necessary precision required for yielding a clear-cut answer to the problem. Even if successful, it would again, for the reason just mentioned, generate (several) minority species that are hard to detect. Therefore, I don't think that this suggestion will be helpful in solving the problem.

Response: We thank your comprehensive review and recognition of our points regarding the suggested XPS measurements.

b) Lack of precision of measurements: The authors convincingly show that a three-electrode experiment reproduces the transition from neutral to the oxidized state and the associated change of displacement at which the high conductance can be observed. I agree with the referee that the two-electrode measurements, even after calibration, are not as precise as those with three or four electrodes, but the authors demonstrate that the basic qualitative behavior is the same, independent of the setup used. Since this new result adds to the reliability of the data analysis, I suggest that these data will be added to the supplement.

Response: Thank you for suggesting. We have included the three-electrode STM-BJ measurement results in the supplementary information.

c) Flicker noise data were extended to the 1 to 10 kHz range and shown to be, within error bars, independent of the range of frequencies used. Therefore, I consider this analysis to be valid.

d) Influence of polarizability of the surrounding electrolyte molecules. This discussion resulted in the correction of an error in the manuscript, but otherwise no new aspects were raised.

Therefore, with the small addition to the supplement, I recommend publication of this paper as is.

Response: We appreciate your acknowledgment and recommending publication.